# Adversarial Score identity Distillation: Rapidly Surpassing the Teacher in One Step

**Mingyuan Zhou**[1,2,*], **Huangjie Zheng**[1], **Yi Gu**[1], **Zhendong Wang**[1], and **Hai Huang**[3,*]

[1]The University of Texas at Austin, [2]Google DeepMind, and [3]Atlassian

## Abstract

Score identity Distillation (SiD) is a data-free method that has achieved state-of-the-art performance in image generation by leveraging only a pretrained diffusion model, without requiring any training data. However, the ultimate performance of SiD is constrained by the accuracy with which the pretrained model captures the true data scores at different stages of the diffusion process. In this paper, we introduce SiDA (SiD with Adversarial Loss), which not only enhances generation quality but also improves distillation efficiency by incorporating real images and adversarial loss. SiDA utilizes the encoder from the generator's score network as a discriminator, allowing it to distinguish between real images and those generated by SiD. The adversarial loss is batch-normalized within each GPU and then combined with the original SiD loss. This integration effectively incorporates the average "fakeness" per GPU batch into the pixel-based SiD loss, enabling SiDA to distill a single-step generator. SiDA converges significantly faster than its predecessor when distilled from scratch, and swiftly improves upon the original model's performance during fine-tuning from a pre-distilled SiD generator. This one-step adversarial distillation method establishes new benchmarks in generation performance when distilling EDM diffusion models, achieving FID scores of **1.499** on CIFAR-10 unconditional, **1.396** on CIFAR-10 conditional, and **1.110** on ImageNet 64x64. When distilling EDM2 models trained on ImageNet 512x512, our SiDA method surpasses even the largest teacher model, EDM2-XXL, which achieved an FID of 1.81 using classifier-free guidance (CFG) and 63 generation steps. Specifically, SiDA achieves FID scores of **2.156** for size XS, **1.669** for S, **1.488** for M, **1.413** for L, **1.379** for XL, and **1.366** for XXL, all without CFG and in a single generation step. These results highlight substantial improvements across all model sizes. Our code and checkpoints are available at https://github.com/mingyuanzhou/SiD/tree/sida.

## 1 Introduction

Modeling the distribution of high-dimensional data, such as natural images, has been a persistent challenge in machine learning (Bishop, 2006; Murphy, 2012; Goodfellow et al., 2016). Before the emergence of deep generative models, research focused primarily on constructing hierarchical models constrained by parametric distributions (Blei et al., 2003; Griffiths & Ghahramani, 2005; Fei-Fei & Perona, 2005; Chong et al., 2009; Zhou et al., 2009) and developing neural networks with stochastic binary hidden layers (Hinton et al., 2006; Salakhutdinov & Hinton, 2009; Vincent et al., 2010), supported by robust inference techniques such as Gibbs sampling, maximum likelihood, variational inference (Hoffman et al., 2013; Blei & Jordan, 2006), and contrastive divergence (Hinton, 2002).

The last decade has witnessed significant advancements in deep generative models, including generative adversarial networks (GANs) (Goodfellow et al., 2014; Reed et al., 2016; Karras et al., 2019), normalizing flows (Papamakarios et al., 2019), and variational auto-encoders (VAEs) (Kingma & Welling, 2014; Rezende et al., 2014). Although VAEs are noted for their stability, they often produce blurred images, while GANs are acclaimed for their ability to generate photorealistic images despite challenges with training instability and generation diversity. These dynamics have driven research towards refining statistical distances to more

---

[*]The majority of the work was done while the authors were at Google.

accurately measure discrepancies between true and generated data distributions, especially in scenarios where traditional metrics fail due to non-overlapping supports in high-dimensional spaces (Arjovsky et al., 2017; Li et al., 2017; Zheng & Zhou, 2021). Diffusion models (Sohl-Dickstein et al., 2015; Song & Ermon, 2019; Ho et al., 2020) have further revolutionized this field with their ability to produce photorealistic images, albeit with multiple iterative refinement resulting in slow sampling speeds. In response, significant efforts have been directed towards developing efficient methods to utilize pretrained diffusion models for reversing the forward diffusion process through iterative refinement.

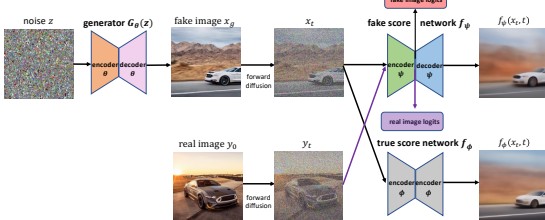

Figure 1: Illustration of the SiDA algorithm. The generator loss, as defined defined in (8), is determined by fake images $\boldsymbol{x}_g$, true-score-net-denoised fake images $f_\phi(\boldsymbol{x}_t, t)$, fake-score-net-denoised fake images $f_\psi(\boldsymbol{x}_t, t)$, and the fake image logits. Meanwhile, the fake score network loss, as defined in (9), is determined by $f_\psi(\boldsymbol{x}_t, t)$ and the fake-score-net encoded logits of both fake and real images.

Departing from conventional methods, Diffusion GAN (Wang et al., 2023b) reframes forward diffusion as a cost-effective, domain-agnostic data-augmentation strategy to tackle issues of non-overlapping distribution support. This approach introduces noise at various levels and utilizes statistical distances such as Jensen–Shannon divergence (JSD) for comparing distributions. The method promotes alignment between the true and generated data distributions at any stage during the forward diffusion process. It advocates for directly guiding the learning of a one-step generator through forward diffusion, moving away from the traditional dependence on iterative refinement-based sampling seen in reverse diffusion processes.

Expanding on this approach, recent methodologies have evolved from directly comparing empirical distributions of diffused groundtruth and generated data via JSD to distill their scores—a notable strength of diffusion models—at various stages of the forward diffusion process. This evolution has catalyzed the development of cutting-edge diffusion distillation techniques such as score distillation sampling (Poole et al., 2023), variational score distillation (Wang et al., 2023c), Diff-Instruct (Luo et al., 2023b), SwiftBrush (Nguyen & Tran, 2024), and distribution matching distillation (DMD) (Yin et al., 2024b). These methods refine models by analyzing the diffused KL divergence between corrupted data and model distributions, effectively leveraging the gradients of this divergence. Although directly computing this KL divergence presents significant challenges, its gradients can be effectively estimated using the scores from both the pretrained model and the current generator.

In pursuit of transcending traditional challenges associated with JSD and model-based KL divergence—which are often difficult to optimize and prone to mode-seeking behaviors, respectively—Zhou et al. (2024b) have pioneered a model-based Fisher divergence within the diffusion distillation framework. This novel approach, notably data-free, has demonstrated for the first time the potential to match or even surpass the performance of teacher models in a single generation step. However, SiD and other data-free methods are based on the assumption that the score produced by teacher networks can well represent the data score. This assumption can potentially create a performance bottleneck for distilled single-step generators, especially if the teacher diffusion model is not well-trained or has limited capacity. In this paper, we aim to further enhance SiD by integrating real-world data. This integration seeks to compensate for the teacher model's limitations in accurately representing the true score, thereby producing even more realistic generative outcomes. This enhancement broadens the practical applications of diffusion distillation in real-world scenarios.

As illustrated in Figure 1, the proposed SiDA algorithm builds on the existing fake score network and generator design of SiD, incorporating a Diffusion GAN-based adversarial loss that discriminates whether generated images are real or fake at various noisy stages of the diffusion process. The discriminator shares the weight with the encoder modules of the fake score network. To ensure compatibility with SiD's pixel-level loss, we compute the discriminator loss at each spatial position of the latent encoder feature map and average across the batch within each GPU. This measure is integrated into the SiD loss for joint distillation and adversarial training and introduces no additional parameters. Our findings across teacher models of various sizes and diverse datasets indicate that SiDA improves iteration efficiency by nearly an order of magnitude when training models initialized with the teacher. Additionally, it achieves remarkably low FID when initialized from the SiD-distilled checkpoints, consistently outperforming the teacher models across all tested datasets and model sizes.

## 2 RELATED WORK

Diffusion models (Song & Ermon, 2019; Ho et al., 2020; Karras et al., 2022) are a family of generative models that adopts a denoising score matching objective (Vincent, 2011; Sohl-Dickstein et al., 2015) to learn the score function of complex high-dimensional distributions at different noise levels. A deep neural network is trained to match the score function of noise-corrupted data via minimizing a data-based Fisher divergence (Song & Ermon, 2019). This trained score network is then used to generate new samples by iteratively denoising random noise. Diffusion models have gained significant attention and success in generative modeling due to training stability and high-quality sample generation, with example applications in text-to-image generation (Nichol et al., 2022; Ramesh et al., 2022; Saharia et al., 2022; Rombach et al., 2022; Podell et al., 2024). However, the sampling process in diffusion models involves gradual denoising across many steps, making it significantly slower compared to one-step generation models like GANs and VAEs.

Numerous techniques have been proposed to accelerate diffusion models (Zheng et al., 2022; Lyu et al., 2022; Luhman & Luhman, 2021; Salimans & Ho, 2022; Zheng et al., 2023; Meng et al., 2023; Kim et al., 2023). Recently, one-step and few-step diffusion distillation methods have gained prominence, achieving the speed of GANs while retaining the robust capabilities of diffusion models. These methods can be broadly classified into two categories: data-free (Luo et al., 2023b; Nguyen & Tran, 2024; Zhou et al., 2024b) and those requiring real images or teacher-synthesized noise-image pairs (Liu et al., 2023; Song & Dhariwal, 2023; Luo et al., 2023a; Kim et al., 2023; Sauer et al., 2023; Xu et al., 2023; Yin et al., 2024b). The latter frequently incorporate GAN-based adversarial learning as a critical component to enhance the generation performance of baseline distillation methods, which vary from consistency models (Song et al., 2023) to score distillation sampling (Poole et al., 2023; Wang et al., 2023c). Without the adversarial component, performance typically declines significantly (Kim et al., 2023; Sauer et al., 2023; Yin et al., 2024a), which raises concerns about the robustness and effectiveness of the underlying base models lacking adversarial learning support. We note that the term "One-Step" in this paper differs from its usage in OSGAN (Shen et al., 2021), which refers to a non-alternating GAN training technique, not a single generator pass.

The collaborative integration of GANs and diffusion models, aimed at leveraging the strengths of both, is not a concept that originated with adversarial distillation. This approach is rooted in earlier studies: Xiao et al. (2022) developed a method to learn the reverse process of the diffusion chain using a time-step-conditioned discriminator. Simultaneously, Zheng et al. (2022) proposed that GANs learn a noisy marginal distribution at a specific targeted timestep, which would then serve as the prior for initiating the reverse diffusion process. Furthermore, they fine-tuned an early version of the Stable Diffusion framework to produce this noisy marginal using a discriminator equipped with an MLP head atop the encoder module of the U-Net in Stable Diffusion, specifically designed to accelerate text-to-image generation in just a few steps while maintaining high quality. A key insight from Zheng et al. (2022) is that the diffusion U-Net backbone can simultaneously be used for reverse diffusion and function as both a Diffusion GAN's generator and discriminator. Building on this, we aim to generalize and simplify the designs of Zheng et al. (2022) by repurposing the score networks in SiD to perform both distillation and adversarial learning, without introducing any additional parameters.

Our work also builds on previous efforts to enhance distribution matching by injecting noise. Measuring differences between distributions in high dimensions, where supports often do not overlap, is challenging and has spurred the development of advanced statistical distances (Murphy, 2012; Arjovsky et al., 2017; Li et al., 2017; Zheng & Zhou, 2021). Adding noise has been shown to be effective in overlapping the supports of the true and fake data distributions, which typically reside in separate high-dimensional density regions (Wang et al., 2023b). This overlap mitigates the limitations of many existing statistical distances in measuring differences between such distributions accurately.

Diffusion GAN introduced a novel strategy that matches the distribution between true and generated data at various stages of the forward diffusion process. This corrupts the observed data at different signal-to-noise ratios and aligns the distributions at multiple points throughout the diffusion pathway, using JSD for comparison to ensure a close alignment of the model and data distributions.

This approach to noise injection-based distribution matching has been further advanced by integrating pretrained diffusion models, which excel at estimating the true data score throughout the forward diffusion process. Building on this, while the KL divergence from diffused true to fake data distributions is intractable to estimate directly, its gradient can be computed using the scores of the true

and fake data at various noise levels (Poole et al., 2023; Wang et al., 2023c). This advancement has driven several recent innovations in one-step diffusion distillation methods for image generation, exemplified by studies such as Luo et al. (2023b), Nguyen & Tran (2024), and Yin et al. (2024b).

## 3 PRELIMINARIES

Below, we briefly review both SiD and Diffusion GAN, the foundations on which SiDA is built. A detailed description of SiDA follows in the next section.

**Score identity Distillation (SiD):** Moving beyond the challenges of computing JSD between diffused empirical data distributions, which can be unstable to optimize, and estimating the gradient of their KL divergence—known for mode-seeking behaviors when expectations are taken with respect to the model distribution—SiD employs a model-based Fisher divergence to match their scores. More specifically, let us denote the forward transition kernel used in Gaussian diffusion models by $q(\boldsymbol{x}_t \,|\, \boldsymbol{x}_0) = \mathcal{N}(a_t \boldsymbol{x}_0, \sigma_t^2 \mathbf{I})$, where $a_t \in (0, 1)$, $\sigma_t > 0$, and the signal-to-noise ratio $a_t^2/\sigma_t^2$ monotonically decreases towards zero as $t$ progresses. SiD explicitly represents the forward diffused data and model distributions as semi-implicit distributions (Yin & Zhou, 2018; Yu et al., 2023b):

$$p_{\text{data}}(\boldsymbol{x}_t) = \int q(\boldsymbol{x}_t|\boldsymbol{x}_0)p_{\text{data}}(\boldsymbol{x}_0)d\boldsymbol{x}_0; \quad p_\theta(\boldsymbol{x}_t) = \int q(\boldsymbol{x}_t|\boldsymbol{x}_g)p_\theta(\boldsymbol{x}_g)d\boldsymbol{x}_g.$$

These distributions, while lacking analytic density functions, are simple to sample from. Specifically, by defining $G_\theta$ as a generator converting noise into data, a diffused fake data $\boldsymbol{x}_t$ can be produced as:

$$\boldsymbol{x}_t = a_t \boldsymbol{x}_g + \sigma_t \boldsymbol{\epsilon}_t, \quad \boldsymbol{\epsilon}_t \sim \mathcal{N}(0, \mathbf{I}), \quad \boldsymbol{x}_g = G_\theta(\boldsymbol{z}, \boldsymbol{c}), \quad \boldsymbol{z} \sim p(\boldsymbol{z}). \tag{1}$$

To distill a student generative model, the distribution $p_\theta(\boldsymbol{x}_t)$ needs to match the noisy data distribution at any time point $t$, ensuring that the distributions $p_{\text{data}}(\boldsymbol{x}_0)$ and $p_\theta(\boldsymbol{x}_g)$ are identical.

To achieve this goal, the distillation process involves adjusting the student model's parameters so that its output at various stages of diffusion aligns closely with the diffused data from the target distribution. This alignment is typically facilitated through optimization techniques that minimize a specific statistical distance between these distributions across the entire diffusion trajectory (Wang et al., 2023b; Luo et al., 2023b). Following the same principle, SiD aims to minimize a model-based Fisher divergence given by:

$$\mathcal{L}_\theta(\phi^*, \psi^*(\theta), t) = \mathbb{E}_{\boldsymbol{x}_t \sim p_\theta(\boldsymbol{x}_t)} \left[ \| \nabla_{\boldsymbol{x}_t} \ln p_{\text{data}}(\boldsymbol{x}_t) - \nabla_{\boldsymbol{x}_t} \ln p_\theta(\boldsymbol{x}_t) \|_2^2 \right]$$

$$= \mathbb{E}_{\boldsymbol{x}_t \sim p_\theta(\boldsymbol{x}_t)} \left[ \frac{a_t^2}{\sigma_t^4} \| f_{\phi^*}(\boldsymbol{x}_t, t) - f_{\psi^*(\theta)}(\boldsymbol{x}_t, t) \|_2^2 \right], \tag{2}$$

where $f_{\phi^*}$ and $f_{\psi^*(\theta)}$ represent the optimal denoising score networks for the true and fake data distributions, respectively. These can be expressed as:

$$f_{\phi^*}(\boldsymbol{x}_t, t) = (\boldsymbol{x}_t + \sigma_t^2 \nabla_{\boldsymbol{x}_t} \ln p_{\text{data}}(\boldsymbol{x}_t))/a_t = \mathbb{E}[\boldsymbol{x}_0 \,|\, \boldsymbol{x}_t],$$

$$f_{\psi^*(\theta)}(\boldsymbol{x}_t, t) = (\boldsymbol{x}_t + \sigma_t^2 \nabla_{\boldsymbol{x}_t} \ln p_\theta(\boldsymbol{x}_t))/a_t = \mathbb{E}[\boldsymbol{x}_g \,|\, \boldsymbol{x}_t]. \tag{3}$$

**GANs and Diffusion GANs:** GANs (Goodfellow et al., 2014) seek to match the distributions with a min-max game between a discriminator $D$ and a generator $G$, with the training loss formulated as:

$$\min_G \max_D \mathbb{E}_{\boldsymbol{x} \sim p_{\text{data}}(\boldsymbol{x})}[\log D(\boldsymbol{x})] + \mathbb{E}_{\boldsymbol{z} \sim p(\boldsymbol{z})}[\log(1 - D(G(\boldsymbol{z})))], \quad \boldsymbol{z} \sim \mathcal{N}(0, I).$$

To overcome the issues of non-overlapping probability support between two distributions, Wang et al. (2023b) propose Diffusion GANs to match the noisy distributions $p_{\text{data}}(\boldsymbol{x}_t)$ and $p_\theta(\boldsymbol{x}_t)$ with a discriminator conditioned on time-step $t$, which is sampled from an adaptive proposal distribution based on the competition between the generator and discriminator. Diffusion GANs are trained with:

$$\min_G \max_D \mathbb{E}_{\boldsymbol{x}_0 \sim p(\boldsymbol{x}_0), \boldsymbol{x}_t \sim q(\boldsymbol{x}_t \,|\, \boldsymbol{x}_0; t)}[\log(D(\boldsymbol{x}_t; t))] + \mathbb{E}_{\boldsymbol{z} \sim p(\boldsymbol{z}), \boldsymbol{x}_t^g \sim q(\boldsymbol{x}_t \,|\, G(\boldsymbol{z}); t)}[\log(1 - D(\boldsymbol{x}_t^g; t))].$$

From the observations above, we recognize that the core objective of both SiD and Diffusion GAN is to align the noisy distributions at any time step $t$ to ensure that the generative distribution $p_\theta(\boldsymbol{x}_g)$ closely matches the clean data distribution $p_{\text{data}}(\boldsymbol{x}_0)$. SiD operates under the assumption that it has access to the true data score $f_{\phi^*}$ (or a pretrained estimation thereof) but not to the actual data, while Diffusion GAN operates under the reverse assumption—access to actual data but not to $f_{\phi^*}$. In the following section, we will detail how these two objective functions are integrated for joint distillation and adversarial generation, leveraging their respective strengths to enhance model performance.

## 4  SIDA: ADVERSARIAL SCORE IDENTITY DISTILLATION

**Motivations for Integrating Adversarial Loss into SiD.**  The model-based Fisher divergence, as shown in (2), is often intractable because neither $\phi^*$ nor $\psi^*(\theta)$ are directly known. A standard initial approximation, common across nearly all diffusion distillation methods, is to replace $f_{\phi^*}$ with the denoising score network $f_\phi$, which is pretrained on data $\boldsymbol{x}_t \sim p_{\text{data}}(\boldsymbol{x}_t)$ and referred to as the teacher. This approach hinges on the assumption that the teacher's score accurately represents the true data score, which could potentially create a performance bottleneck for distilled generators. Essentially, as the pretrained teacher $f_\phi$ does not perfectly recover the true data score $f_{\phi^*}$, the objective function:

$$\mathcal{L}_\theta(\boldsymbol{x}_t, \phi, \psi^*(\theta)) = \mathbb{E}_{\boldsymbol{x}_t \sim p_\theta(\boldsymbol{x}_t)} \left[ \tfrac{a_t^2}{\sigma_t^4} \| f_\phi(\boldsymbol{x}_t, t) - f_{\psi^*(\theta)}(\boldsymbol{x}_t, t) \|_2^2 \right], \tag{4}$$

represents only an approximation of (2), meaning $\mathcal{L}_\theta(\boldsymbol{x}_t, \phi, \psi^*(\theta)) \neq \mathcal{L}_\theta(\boldsymbol{x}_t, \phi^*, \psi^*(\theta))$. Optimizing this function may not ensure that the distribution $p_\theta(\boldsymbol{x}_g)$ matches $p_{\text{data}}(\boldsymbol{x}_0)$.

This discrepancy motivates the integration of Diffusion GAN-based adversarial loss to potentially correct deviations caused by inaccuracies in $f_\phi$, aiming for a closer alignment between the generated and true data distributions. Specifically, while optimizing (4) does not guarantee that $p_\theta(\boldsymbol{x}_g)$ will closely align with $p_{\text{data}}(\boldsymbol{x}_0)$, particularly if $f_\phi$ is not well-trained or has limited capacity, incorporating a Diffusion GAN-based adversarial loss can enhance the match between the distributions of diffused true data and diffused generator-synthesized fake data, thus improving the correspondence between $p_\theta(\boldsymbol{x}_g)$ and $p_{\text{data}}(\boldsymbol{x}_0)$. Although introducing a separate discriminator to implement the adversarial loss is possible, we find it unnecessary as we can effectively reuse the fake score network, thereby avoiding the introduction of any new parameters. This approach simplifies the model architecture and enhances efficiency, as further explained below.

**Joint Score Estimation and Discrimination.**  While $\psi^*(\theta)$ is intractable and would introduce a complex bi-level optimization problem due to its dependence on $\theta$—the very parameter we aim to optimize—SiD acknowledges the impracticality of obtaining either $\psi^*(\theta)$ or its gradient with respect to $\theta$. Therefore, the same as many previous works in diffusion distillation, SiD estimates $\psi^*(\theta)$ using a denoising network $\psi$, which is learned on the fake data, as

$$\min_\psi \hat{\mathcal{L}}_\psi(\boldsymbol{x}_t, t) = \gamma(t) \| f_\psi(\boldsymbol{x}_t, t) - \boldsymbol{x}_g \|_2^2, \tag{5}$$

where $\boldsymbol{x}_t$ and $\boldsymbol{x}_g$ are drawn via reparameterization, as in (1), and $\gamma(t)$ is a reweighting term that is typically set the same as the signal-to-noise ratio at time $t$ (Ho et al., 2020; Kingma et al., 2021; Hang et al., 2023). This approximation introduces bias, which we will address and demonstrate how to mitigate effectively in later discussion.

To avoid introducing any additional parameters or complex training pipelines, we incorporate a return-flag as an additional input to the network $f_\psi$, offering options 'decoder', 'encoder', and 'encoder-decoder'. When set to 'decoder', the network returns the denoised image as before. If set to 'encoder', it performs average pooling on the output of the last U-Net encoder block of $f_\psi$ along the channel dimension and returns a 2D discriminator map. For 'encoder-decoder', the network outputs both the discriminator map and continues through the U-Net decoder blocks to return the denoised image. The choice of option depends on whether we are updating the fake score network or the generator, and whether the input is a real image or a generator-synthesized image. With the capability to extract 2D discriminator maps either under the 'encoder' option or jointly with the denoised images under the 'encoder-decoder' option, we are now positioned to define the adversarial loss function.

The gradient of this loss function will either directly impact the encoder part of $f_\psi$ or the generator, contingent upon the optimization target—whether it is the joint loss involving fake-score estimation and discrimination or the combined loss of diffusion distillation and adversarial generation. It is also important to highlight that no additional parameters have been introduced to facilitate the joint estimation of fake scores and discrimination between true and fake images. This streamlined approach aids in maintaining model efficiency without compromising the effectiveness of the methodology.

**Joint Diffusion Distillation and Adversarial Generation.**  While recognizing the impracticality of directly obtaining either $\psi^*(\theta)$ or its gradient with respect to $\theta$, substituting $\psi^*(\theta)$ in (4) with a denoising network $\psi$ could introduce a severely biased gradient that undermines the optimization of $\theta$. SiD addresses this challenge in gradient estimation by subtracting a biased gradient estimate, which has proven ineffective in practice, from a less biased one that has shown standalone effectiveness.

This strategy results in a loss function that significantly enhances performance, expressed as:

$$\tilde{\mathcal{L}}_{\theta,t,\alpha} = \omega(t)\frac{a_t^2}{\sigma_t^4}\mathcal{L}_{\theta,t,\alpha}^{(\text{sid})},\tag{6}$$

where $\omega(t)$ are weight coefficients, $\alpha$ is a bias correction weight that is typically set at 1 or 1.2, and

$$\mathcal{L}_{\theta,t,\alpha}^{(\text{sid})} = -\alpha\|f_\phi(\boldsymbol{x}_t,t) - f_\psi(\boldsymbol{x}_t,t)\|_2^2 + (f_\phi(\boldsymbol{x}_t,t) - f_\psi(\boldsymbol{x}_t,t))^T(f_\phi(\boldsymbol{x}_t,t) - \boldsymbol{x}_g).$$

SiD employs an alternating optimization strategy, updating $\psi$ according to (5) and $\theta$ according to (6), using only generator-synthesized fake data, drawn as specified in (1), for parameter updates. This approach ensures that both components are iteratively refined with outputs generated by the model itself, without relying on real data inputs.

An alternative theoretical justification for the SiD loss presented in (6) has been recently explored by Luo et al. (2024), demonstrating that this loss, when $\alpha = 1$, can be derived under certain assumptions about the data generation mechanism, specifically involving stop-gradient conditions. However, regardless of the method intended to recover the true theoretical loss that would yield the SiD gradient used in practice, it is essential to recognize that, as long as $\psi^*(\theta)$ or its gradient-detached version is part of the intended theoretical loss formulation—including model-based Fisher divergence—the bias introduced by replacing $\psi^*(\theta)$ with $\psi$ remains generally unavoidable in practice. This highlights the importance of empirical validation to ensure that the bias correction strategy in place is effective.

Since $f_\phi$ has been adapted to serve dual purposes—not only performing denoising but also capable of outputting a 2D discriminator map for the encoder's latent space—we are now ready to define an adversarial generation loss. This loss compels the generator to trick the discriminator map into believing its output originates from a diffused real image at each spatial location of this latent 2D discriminator map. Consequently, the generator will be updated using both the loss given by (6) and a Diffusion GAN's generator loss, effectively integrating these components to bridge the gap between the pretrained score $f_\phi$ and the true score $f_\phi^*$, and hence enhance the realism of the generated outputs.

**Loss Construction that Respects the Weighting of Diffusion Distillation Loss.** Building on the discussions above, we are now prepared to synthesize all components into a coherent loss function. We face an additional challenge: the diffusion loss used by SiD operates at the pixel level, while the Diffusion GAN loss functions at the level of the latent 2D space. This discrepancy can make it challenging to balance the losses between diffusion distillation and adversarial generation. To circumvent the need for dataset-specific adjustments and to develop a universally applicable solution, we have devised a strategy that honors the existing weighting mechanisms used in training and distilling diffusion models. This approach involves performing GPU batch pooling to calculate an average "fakeness" within each GPU batch, then incorporating this average fakeness into each pixel's loss prior to applying the pixel-specific weights. We have found this per-GPU-batch fakeness strategy to be highly effective, demonstrating robust performance across all datasets considered in this study.

More specifically, denoting $(W, H, C_c)$ as the size of the image, $\mathcal{B}$ a batch inside a GPU, $|\mathcal{B}|$ the per GPU batch size, $D$ the encoder part of $f_\psi$, $D(\boldsymbol{x}_t)$ the 2D discriminator map of size $(W', H')$ given input $\boldsymbol{x}_t$, and $D(\boldsymbol{x}_t)[i,j]$ the $(i,j)th$ element of $D(\boldsymbol{x}_t)$, we define the adversarial loss that measures the average in GPU-batch fakeness as

$$\mathcal{L}_\theta^{(\text{adv})} = \frac{1}{|\mathcal{B}|}\sum_{\boldsymbol{x}_t \in \mathcal{B}}\ln D(\boldsymbol{x}_t) = \frac{1}{|\mathcal{B}|W'H'}\sum_{\boldsymbol{x}_t \in \mathcal{B}}\sum_{i=1}^{W'}\sum_{j=1}^{H'}\ln D(\boldsymbol{x}_t)[i,j].\tag{7}$$

We now define SiDA's generator loss, aggregated over a GPU batch, as follows:

$$\mathcal{L}_\theta^{(\text{sida})}(\mathcal{B}) = \sum_{\boldsymbol{x}_t \in \mathcal{B}}\frac{\omega(t)a_t^2}{2\sigma_t^4}\left(\lambda_{\text{sid}}\mathcal{L}_{\theta,t}^{(\text{sid})} + \lambda_{\text{adv},\theta}C_cWH\mathcal{L}_\theta^{(\text{adv})}\right),\tag{8}$$

where we set $\lambda_{\text{sid}} = 100$ and $\lambda_{\text{adv},\theta} = 0.01$ by default, unless specified otherwise. We note that while $\mathcal{L}_{\theta,t}^{(\text{sid})}$ is influenced by the discrepancy between the teacher network $\phi$ and its optimal value $\phi^*$, $\mathcal{L}_\theta^{(\text{adv})}$ interacts directly with the training data, helping to mitigate this discrepancy. For the fake score network $f_\psi$, whose encoder also serves as the discriminator $D$, we define the SiDA's loss aggregated over a GPU batch as

$$L_\psi^{(\text{sida})} = \sum_{\boldsymbol{x}_t \in \mathcal{B}}\gamma(t)(\|f_\psi(\boldsymbol{x}_t,t) - \boldsymbol{x}_g\|_2^2 + \lambda_{\text{adv},\psi}L_\psi^{(\text{adv})}),\tag{9}$$

where the discriminator loss given a batch of fake data $\boldsymbol{x}_t$ and real data $\boldsymbol{y}_t$ is expressed as

$$L_\psi^{(\text{adv})} = \frac{1}{2|\mathcal{B}|W'H'}\sum_{\boldsymbol{y}_t,\boldsymbol{x}_t \in \mathcal{B}}\sum_{i'=1}^{W'}\sum_{j'=1}^{H'}\ln(D(\boldsymbol{y}_t)[i',j']) + \ln(1 - D(\boldsymbol{x}_t)[i',j']).\tag{10}$$

Here, $\lambda_{\text{adv},\psi}$ is a hyperparameter that we adjust to approximately balance the loss scales of both terms in the summation shown in (9). Specifically, we set $\lambda_{\text{adv},\psi} = 1$ for distilling all EDM-pretrained models and $\lambda_{\text{adv},\psi} = 100$ for distilling all EDM2-pretrained models.

We provide an overview of the SiDA implementation in Algorithm 1, which begins training from a pretrained EDM diffusion model. In Algorithm 2, we outline the modifications specific to EDM2, and in Algorithm 3, we present the SiDA algorithm that initiates training from both a pretrained diffusion model and a pre-distilled SiD generator, referred to as SiD$^2$A (SiD-initialized SiDA).

## 5    EXPERIMENTAL RESULTS

We evaluate SiDA through comprehensive experiments designed to assess its effectiveness compared to the teacher diffusion model and other existing baseline methods for both unconditional and label-conditional image generation. We present results that measure both quantitative and qualitative performance across various datasets, aiming to demonstrate SiDA's capability to generate high-quality samples through efficient score distillation.

We focus on distilling EDM (Karras et al., 2022) diffusion models pretrained on four distinct datasets, as well as six different-sized EDM2 (Karras et al., 2024), all pretrained on ImageNet $512 \times 512$. Notably, while the SiD technique (Zhou et al., 2024b) has been evaluated using EDM, it has not yet been tested on EDM2. This paper makes a supplementary contribution by adapting the SiD codebase for compatibility with the EDM2 framework, which represents an advanced evolution of EDM, and establishing it as a stong baseline for comparison. For detailed descriptions of the experimental setup, datasets, and evaluation metrics, please refer to Appendix A. For details of the algorithm and modifications needed to adapt SiD and SiDA from EDM to EDM2, prlease refer to Appendix B.

As outlined in Tables 6 and 7, SiDA introduces negligible memory overhead compared to SiD, owing to its design that introduces no additional parameters. Although SiDA experiences an approximate 10% reduction in iteration speed and requires real data, it converges significantly faster than SiD and consistently delivers superior performance relative to the teacher model, as demonstrated below.

### 5.1    ABLATION ON THE CHOICE OF $\alpha$.

The gradient bias correction weight factor $\alpha$ is an important hyperparameter in SiD, typically set to 1 or 1.2. We conducted an ablation study to assess its impact on the performance of SiDA. Specifically, we tested a spectrum of $\alpha$ values $[-0.25, 0.0, 0.5, 0.75, 1.0, 1.2, 1.5]$ during the distillation of the EDM model pretrained on CIFAR-10 (unconditional). Figure 2 demonstrates how the model's performance varies with different $\alpha$ values over the first 20k iterations at a batch size of 256, assessed using FID and IS. We determined that an $\alpha$ value of 1.0 yielded the best performance, prompting us to adopt this setting for subsequent experiments of SiDA over various datasets. In contrast, lower $\alpha$ values such as $-0.25$ or 0 resulted in suboptimal performance, while high values like 1.2 or 1.5 led to training instability and divergence. Interestingly, the performance disparity for $\alpha$ values between 0.5 and 1.0 was significantly narrower than those observed in SiD, as depicted in Figure 8 of Zhou et al. (2024b). These findings imply that although the adversarial

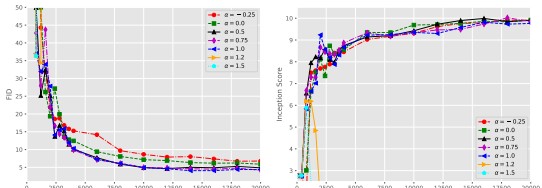

Figure 2: Ablation Study of $\alpha$ on distilling an EDM model pretrained on CIFAR-10 (unconditional): Each plot illustrates the relation between the performance, measured by FID (*left*) and Inception Score (*right*) *vs.* the number of training iterations during distillation, across varying values of $\alpha$. The batch sizer is 256. The study underscores the impact of $\alpha$ on both training efficiency and generative fidelity, leading us to select $\alpha = 1.0$ for subsequent experiments.

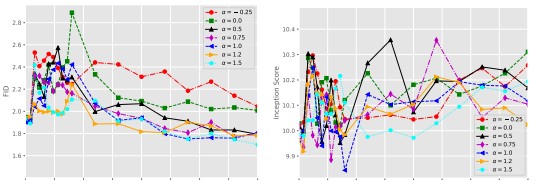

Figure 3: Analogous plot for SiD$^2$A, a SiD initialized SiDA model. We observe a trajectory of improvements in SiD$^2$A following an initial warmup period, and more robust performance relative to $\alpha$ choice.

loss may introduce some instability into the distillation process, it could also compensate for the inadequate correction of gradient biases associated with smaller $\alpha$ values.

| Family | Model | NFE | FID (↓) | IS (↑) |
|---|---|---|---|---|
| Teacher | VP-EDM (Karras et al., 2022) | 35 | 1.97 | 9.68 |
| Diffusion | DDPM (Ho et al., 2020) | 1000 | 3.17 | 9.46 ± 0.11 |
| | DDIM (Song et al., 2020) | 100 | 4.16 | |
| | DPM-Solver-3 (Lu et al., 2022) | 48 | 2.65 | |
| | VDM (Kingma et al., 2021) | 1000 | 4.00 | |
| | iDDPM (Nichol & Dhariwal, 2021) | 4000 | 2.90 | |
| | HSIVI-SM (Yu et al., 2023b) | 15 | 4.17 | |
| | TDPM+ (Zheng et al., 2022) | 100 | 2.83 | 9.34 |
| | VP-EDM+LEGO-PR (Zheng et al., 2024b) | 35 | 1.88 | 9.84 |
| One Step | StyleGAN2+ADA+Tune (Karras et al., 2020) | 1 | 2.92 ± 0.05 | 9.83 ± 0.04 |
| | Diffusion ProjectedGAN (Wang et al., 2023b) | 1 | 2.54 | |
| | iCT-deep (Song & Dhariwal, 2023) | 1 | 2.51 | 9.76 |
| | Diff-Instruct (Luo et al., 2023b) | 1 | 4.53 | 9.89 |
| | StyleGAN2+ADA+Tune+DI (Luo et al., 2023b) | 1 | 2.71 | 9.86 ± 0.04 |
| | DMD (Yin et al., 2024b) | 1 | 3.77 | |
| | CTM (Kim et al., 2023) | 1 | 1.98 | |
| | GDD-I (Zheng & Yang, 2024) | 1 | 1.54 | 10.10 |
| | SiD, $\alpha = 1.0$ | 1 | 2.028 ± 0.020 | 10.017 ± 0.047 |
| | SiD, $\alpha = 1.2$ | 1 | 1.923 ± 0.017 | 9.980 ± 0.042 |
| | SiDA, $\alpha = 1.0$ | 1 | 1.516 ± 0.010 | **10.323** ± 0.048 |
| | SiD²A, $\alpha = 1.0$ | 1 | **1.499** ± 0.012 | 10.188 ± 0.035 |
| | SiD²A, $\alpha = 1.2$ | 1 | 1.519 ± 0.009 | 10.252 ± 0.027 |

| Family | Model | FID (↓) |
|---|---|---|
| Teacher | VP-EDM (Karras et al., 2022) | 1.79 |
| Direct generation | BigGAN (Brock et al., 2019) | 14.73 |
| | StyleGAN2+ADA (Karras et al., 2020) | 3.49 ± 0.17 |
| | StyleGAN2+ADA+Tune (Karras et al., 2020) | 2.42 ± 0.04 |
| Distillation | GET-Base (Geng et al., 2023) | 6.25 |
| | Diff-Instruct (Luo et al., 2023b) | 4.19 |
| | StyleGAN2+ADA+Tune+DI (Luo et al., 2023b) | 2.27 |
| | DMD (Yin et al., 2024b) | 2.66 |
| | DMD (w.o. KL) (Yin et al., 2024b) | 3.82 |
| | DMD (w.o. reg.) (Yin et al., 2024b) | 5.58 |
| | CTM (Kim et al., 2023) | 1.73 |
| | GDD-I (Zheng & Yang, 2024) | 1.44 |
| | SiD, $\alpha = 1.0$ | 1.932 ± 0.019 |
| | SiD , $\alpha = 1.2$ | 1.710 ± 0.011 |
| | SiDA, $\alpha = 1.0$ | 1.436 ± 0.009 |
| | SiD²A, $\alpha = 1.0$ | 1.403 ± 0.010 |
| | SiD²A, $\alpha = 1.2$ | **1.396** ± 0.014 |

Table 1: Comparison of unconditional generation on CIFAR-10. The best one/few-step generator under the FID or IS metric is highlighted with **bold**.

Table 2: Analogous to Table 1 for CIFAR-10 (conditional). "Direct generation" and "Distillation" methods presented in the table requires one single NFE, and the teacher requires 35 NFE.

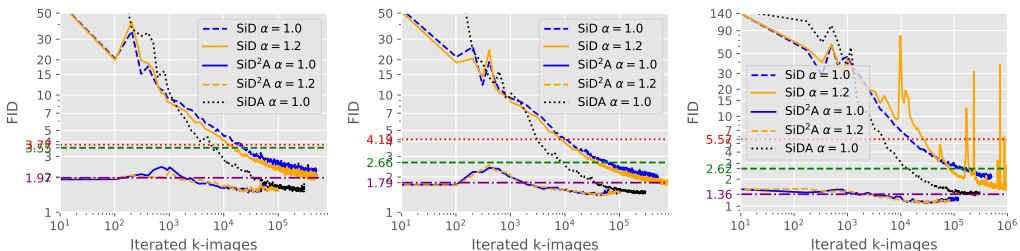

Figure 4: Evolution of FIDs for the SiD and SiDA generator during the distillation of the EDM teacher model pretrained on CIFAR-10 unconditional (*left*) and conditional (*middle*), with a batch size of 256, and on ImageNet 64x64 (*right*) with a batch size of 8192, using $\alpha = 1.0$ or $\alpha = 1.2$. The performance of EDM (35 NFEs), along with DMD and Diff-Instruct, is depicted with horizontal lines in purple, green, and red, respectively.

In an alternative setting, referred to as SiD²A (SiD-initialized SiDA), where the generator $G_\theta$ is initialized using the best available pre-distilled SiD generator, we observe robust and stabilized training across all choices of $\alpha$. This suggests that initializing with the pre-distilled SiD generator provides a more advantageous starting point to achieve enhanced training stability and generative quality. Following these observations, and given that $\alpha = 1.0$ and $\alpha = 1.2$ were also utilized in SiD experiments, we have selected these same values for subsequent experiments with SiD²A for EDM.

## 5.2 BENCHMARKING EDM DISTILLATION

We begin by assessing the performance of SiDA and SiD²A, focusing on both generation quality and convergence speed across different datasets. Our comprehensive evaluation compares SiDA and SiD²A to leading deep generative models, including GANs, diffusion models, their distilled counterparts, and various variations, focusing on generation quality. Qualitatively, random images generated by SiD²A in a single step are displayed from Figures 11 to 15 in the Appendix.

Quantitatively, on CIFAR-10 in both conditional and unconditional settings, SiDA and SiD²A outperform leading models, with details in Tables 1 and 2. Notably, SiDA achieves an FID of 1.516, outperforming all baseline models. When leveraging the one-step generators already distilled by SiD for initialization, SiD²A further advances this performance, achieving an exceptionally low FID of **1.499** on CIFAR-10 unconditional with $\alpha = 1.0$, and **1.396** on CIFAR-10 conditional with $\alpha = 1.2$.

We further validate the effectiveness of SiDA and SiD²A on the ImageNet 64x64 dataset. Here, SiDA demonstrated superior performance over SiD, achieving an FID of 1.353. Notably, by leveraging the advanced capabilities of the SiD one-step generator, SiD²A attained an unprecedented low FID of **1.110** on ImageNet 64x64 with $\alpha = 1.2$, surpassing the teacher model's performance by a large margin. On the FFHQ 64x64 and AFHQ-v2 64x64 datasets, known for human and animal faces, SiDA and SiD²A again prove superior. The datasets' less diverse patterns do not diminish the models' performance, with SiDA and SiD²A maintaining

faster convergence and surpassing the teacher model rapidly. The clear improvements across diverse datasets underscore their effectiveness.

It's worth noting that SiDA's close competitor for ImageNet 64x64, GDD-I (Zheng & Yang, 2024), uses a discriminator inspired by Projected GAN (Sauer et al., 2021), integrating features from VGG16-BN and EfficientNet-lite0. Concerns arose, as discussed in Kynkäänniemi et al. (2023) and Song & Dhariwal (2023), about the potential for encoders pretrained on ImageNet to inadvertently enhance FID scores through feature leakage. This potential issue does not affect SiD, SiDA, or SiD$^2$A, ensuring our results reflect true model performance without such confounds.

SiDA also demonstrates accelerated convergence, as evidenced in Figures 4 and 6. The logarithmic scale plots show SiDA's FID decreasing more rapidly than SiD's, an

Table 3: Analogous to Table 1 for ImageNet 64x64 with label conditioning, using a pretrained EDM diffusion model as the teacher. The Precision and Recall metrics are also included.

| Family | Model | NFE | FID ($\downarrow$) | Prec. ($\uparrow$) | Rec. ($\uparrow$) |
|---|---|---|---|---|---|
| Teacher | VP-EDM (Karras et al., 2022) | 511 | 1.36 | | |
| | | 79 | 2.64 | 0.71 | 0.67 |
| Diffusion | RIN (Jabri et al., 2022) | 1000 | 1.23 | | |
| | DDPM (Ho et al., 2020) | 250 | 11.00 | 0.67 | 0.58 |
| | ADM (Dhariwal & Nichol, 2021a) | 250 | 2.07 | 0.74 | 0.63 |
| | DPM-Solver-3 (Lu et al., 2022) | 50 | 17.52 | | |
| | HSIVI-SM (Yu et al., 2023b) | 15 | 15.49 | | |
| | U-ViT (Bao et al., 2022) | 50 | 4.26 | | |
| | DiT-L/2 (Peebles & Xie, 2023) | 250 | 2.91 | | |
| | LEGO (Zheng et al., 2024b) | 250 | 2.16 | | |
| Consistency | iCT (Song & Dhariwal, 2023) | 1 | 4.02 | 0.70 | 0.63 |
| | iCT-deep (Song & Dhariwal, 2023) | 1 | 3.25 | 0.72 | 0.63 |
| Distillation | PD (Salimans & Ho, 2022) | 2 | 8.95 | 0.63 | **0.65** |
| | G-distill (Meng et al., 2023) ($w$=0.3) | 8 | 2.05 | | |
| | BOOT (Gu et al., 2023) | 1 | 16.3 | 0.68 | 0.36 |
| | PID (Tee et al., 2024) | 1 | 9.49 | | |
| | DFNO (Zheng et al., 2023) | 1 | 7.83 | | 0.61 |
| | CD-LPIPS (Song et al., 2023) | 2 | 4.70 | 0.69 | 0.64 |
| | Diff-Instruct (Luo et al., 2023b) | 1 | 5.57 | | |
| | TRACT (Berthelot et al., 2023) | 2 | 4.97 | | |
| | DMD (Yin et al., 2024b) | 1 | 2.62 | | |
| | CTM (Kim et al., 2023) | 1 | 1.92 | | 0.57 |
| | CTM (Kim et al., 2023) | 2 | 1.73 | | 0.57 |
| | GDD-I (Zheng & Yang, 2024) | 1 | 1.16 | **0.75** | 0.60 |
| | EMD-16 (Xie et al., 2024) | 1 | 2.2 | | 0.59 |
| | DMD2 (Yin et al., 2024a) | 1 | 1.51 | | |
| | DMD2+longer training (Yin et al., 2024a) | 1 | 1.28 | | |
| | SiD , $\alpha = 1.0$ | 1 | $2.022 \pm 0.031$ | 0.73 | 0.63 |
| | SiD , $\alpha = 1.2$ | 1 | $1.524 \pm 0.009$ | 0.74 | 0.63 |
| | SiDA, $\alpha = 1.0$ (ours) | 1 | $1.353 \pm 0.025$ | 0.74 | 0.63 |
| | SiD$^2$A, $\alpha = 1.0$ (ours) | 1 | $1.114 \pm 0.019$ | **0.75** | 0.62 |
| | SiD$^2$A, $\alpha = 1.2$ (ours) | 1 | **1.110** $\pm 0.018$ | **0.75** | 0.62 |

efficiency that also holds when scaled up to ImageNet 64x64. Overall, SiDA consistently outperforms SiD in terms of convergence speed by a substantial margin, marking a significant acceleration, particularly noteworthy since SiD already converges at an exponential rate. This rapid improvement in both speed and performance underscores SiDA's effectiveness and efficiency in generating high-quality images across different datasets. These results not only highlight SiD$^2$A's enhanced image quality and reduced iteration needs but also the overall efficiency of SiDA and SiD$^2$A across various datasets, confirming the advantages of these advanced distillation techniques.

## 5.3 BENCHMARKING EDM2 DISTILLATION

We evaluate the effectiveness of SiD, SiDA, and SiD$^2$A in distilling EDM2 models of six different sizes, all pretrained on ImageNet 512x512. Our findings and comparisons are detailed in Tables 4, 5, and 10 and depicted in Figure 5. The rapid convergence of the algorithms is visually

Table 4: FID scores and number of parameters for various methods on ImageNet 512x512 are presented. The results for EDM2 are sourced from Karras et al. (2024), while those for sCT and sCD are obtained from Lu & Song (2024).

| Method | CFG | NFE | XS (125M) | S (280M) | M (498M) | L (777M) | XL (1.1B) | XXL (1.5B) |
|---|---|---|---|---|---|---|---|---|
| EDM2 | N | 63 | 3.53 | 2.56 | 2.25 | 2.06 | 1.96 | 1.91 |
| EDM2 | Y | 63x2 | 2.91 | 2.23 | 2.01 | 1.88 | 1.85 | 1.81 |
| sCT | Y | 1 | - | 10.13 | 5.84 | 5.15 | 4.33 | 4.29 |
| sCT | Y | 2 | - | 9.86 | 5.53 | 4.65 | 3.73 | 3.76 |
| sCD | Y | 1 | - | 3.07 | 2.75 | 2.55 | 2.40 | 2.28 |
| sCD | Y | 2 | - | 2.50 | 2.26 | 2.04 | 1.93 | 1.88 |
| SiD | N | 1 | $3.353 \pm 0.041$ | $2.707 \pm 0.054$ | $2.060 \pm 0.038$ | $1.907 \pm 0.016$ | $1.888 \pm 0.030$ | $1.969 \pm 0.029$ |
| SiDA | N | 1 | $2.228 \pm 0.037$ | $1.756 \pm 0.023$ | $1.546 \pm 0.023$ | $1.501 \pm 0.024$ | $1.428 \pm 0.018$ | $1.503 \pm 0.020$ |
| SiD$^2$A | N | 1 | **2.156** $\pm 0.028$ | **1.669** $\pm 0.019$ | **1.488** $\pm 0.038$ | **1.413** $\pm 0.022$ | **1.379** $\pm 0.017$ | **1.366** $\pm 0.015$ |

illustrated in Figures 8 through 10. Random images generated by SiD$^2$A in a single step are displayed in Figures 16 through 34. Notably, SiD of Zhou et al. (2024b), the baseline single-step algorithm on which SiDA is based, already performs comparably to the EDM2 teacher, which requires 63 NFEs, and to the stable and scalable consistency distillation (sCD) method of Lu & Song (2024) that uses 2 NFEs—a concurrent development to SiDA. Moreover, SiD convincingly outperforms sCD with 1 NFE, even without using CFG during distillation. As indicated in Table 5, SiD, operating with a single step and without CFG in a data-free setting, ranks among the top performers of all generative models reported on ImageNet 512x512. This state-of-the-art performance provides a robust foundation for both SiDA and SiD$^2$A, which will incorporate real data in their processes.

In line with these distillation results for the EDM family, both SiDA and SiD$^2$A show superior performance over SiD in distilling EDM2, achieving FIDs of 1.756 and 1.669, respectively, with the EDM2-S model, which has only 280M parameters. These scores already outperform the EDM2-XXL model and sCD at 1.5B parameters, whose FIDs exceed 1.8. Remarkably, when scaled to EDM2-XL and EDM2-XXL, SiD$^2$A achieves record-low FIDs of **1.379** and **1.366**, respectively, significantly surpassing previous methods, whether single or multi-step.

Table 5: Performance comparison of generative models trained on ImageNet at 512×512 resolution (Left: without Classifier-Free Guidance (CFG); Right: with CFG). The results for EDM2 of different sizes and their distillations using sCD, SiD, and SiDA algorithms are provided in Table 4.

| Method (CFG=N) | NFE (↓) | #Params | FID (↓) | Method (CFG=Y) | NFE (↓) | #Params | FID (↓) |
|---|---|---|---|---|---|---|---|
| ADM-G (Dhariwal & Nichol, 2021b) | 250 | 559M | 23.24 | ADM-G (Dhariwal & Nichol, 2021b) | 250×2 | >559M | 7.72 |
| U-DiT-B (Tian et al., 2024b) | 250 | 204M | 15.39 | U-ViT-L/4 (Bao et al., 2023) | 250×2 | 287M | 4.67 |
| DiT-XL/2 (Peebles & Xie, 2022) | 250 | 675M | 12.03 | U-ViT-H/4 (Bao et al., 2023) | 250×2 | 501M | 4.05 |
| MaskDiT (Zheng et al., 2024a) | 79 | 736M | 10.79 | ADM-U (Dhariwal & Nichol, 2021b) | 250×2 | >559M | 3.85 |
| ADM-U (Dhariwal & Nichol, 2021b) | 250 | >559M | 9.96 | LEGO-XL-PR (Zheng et al., 2024b) | 250×2 | 681M | 3.99 |
| LEGO-XL-PR (Zheng et al., 2024b) | 250 | 681M | 9.01 | DiM-H (Teng et al., 2024) | 250×2 | 860M | 3.78 |
| BigGAN-deep (Brock et al., 2019) | 1 | 160M | 8.43 | LEGO-XL-PG (Zheng et al., 2024b) | 250×2 | 681M | 3.74 |
| DiMR-XL/3R (Liu et al., 2024) | 250 | 525M | 7.93 | DyDiT-XL (Zhao et al., 2024) | 100×2 | 675M | 3.61 |
| MaskGIT (Chang et al., 2022) | 12 | 227M | 7.32 | DiffuSSM-XL-G (Yan et al., 2024) | 250×2 | 660M | 3.41 |
| MAGVIT-v2 (Yu et al., 2023a) | 12 | 307M | 4.61 | DiT-XL/2 (Peebles & Xie, 2022) | 250×2 | 675M | 3.04 |
| RIN (Jabri et al., 2022) | 1000 | 320M | 3.95 | DRWKV-H/2 (Fei et al., 2024b) | 250×2 | 779M | 2.95 |
| MAGVIT-v2 (Yu et al., 2023a) | 64 | 307M | 3.07 | DiMR-XL/3R (Liu et al., 2024) | 250×2 | 525M | 2.89 |
| VDM++ (Kingma & Gao, 2023) | 512 | 2B | 2.99 | DiS-H/2 (Fei et al., 2024a) | 250×2 | 900M | 2.88 |
| MAR (Li et al., 2024) | 100 | 481M | 2.74 | DiffiT (Hatamizadeh et al., 2025) | 250×2 | 561M | 2.67 |
| StyleGAN-XL (Sauer et al., 2022) | 1×2 | 168M | 2.41 | VDM++ (Kingma & Gao, 2023) | 512×2 | 2B | 2.65 |
| SiD²A-EDM2-XS (ours) | 1 | 125M | 2.156 | VAR-d36-s (Tian et al., 2024a) | 10×2 | 2.3B | 2.63 |
| SiD²A-EDM2-S (ours) | 1 | 280M | 1.669 | SiT-XL (Ma et al., 2024) | 250×2 | 675M | 2.62 |
| SiD²A-EDM2-M (ours) | 1 | 498M | 1.488 | Large-DiT-3B-G[1] (Zhang et al., 2023) | 250×2 | 3B | 2.52 |
| SiD²A-EDM2-L (ours) | 1 | 777M | 1.413 | MaskDiT-G (Zheng et al., 2024a) | 79×2 | 736M | 2.50 |
| SiD²A-EDM2-XL (ours) | 1 | 1.1B | 1.379 | MAGVIT-v2 (Yu et al., 2023a) | 64×2 | 307M | 1.91 |
| SiD²A-EDM2-XXL (ours) | 1 | 1.5B | **1.366** | MAR[2] (Li et al., 2024) | 100×2 | 481M | 1.73 |

(a) EDM2-XS    (b) EDM2-S    (c) EDM2-M    (d) EDM2-L    (e) EDM2-XL    (f) EDM2-XXL

Figure 5: Evolution of FIDs for the SiD, SiDA, and SiD²A generators during the distillation of the EDM2 teacher models of six different sizes pretrained on ImageNet 512x512, with a batch size of 2048 and $\alpha = 1.0$. The performance of EDM2 without classifier-free guidance (CFG) and EDM2 with CFG, using 63 NFEs, along with the simple, stable, and scalable consistency distillation (sCD) method of Lu & Song (2024) with 1 and 2 NFEs, is depicted with horizontal lines in purple, red, green, and cyan respectively.

Mirroring findings from EDM distillation, SiDA also exhibits accelerated convergence when distilling EDM2, as shown in Figure 5. Logarithmic-scale plots show that SiDA achieves an order of magnitude faster decrease in FID compared to SiD, with efficiency that scales effectively to ImageNet 512x512 and across generators of varying sizes. For instance, while SiD required training the generator with approximately 100M generator-synthesized images to match or surpass the teacher's performance, SiDA needed only 10M such images to exceed the teacher's performance. Overall, while SiD proves to be a strong baseline, rivaling the performance of the teacher and methods published subsequently, SiDA consistently surpasses both the teacher and SiD by large margins in terms of final metrics and convergence speed. This rapid improvement underscores SiDA's effectiveness and efficiency in generating high-resolution images in a single step across various model sizes.

## 6 CONCLUSION

In this paper, we present Score identity Distillation with Adversarial Loss (**SiDA**), an innovative framework that optimizes diffusion model distillation by integrating the strengths of Score identity Distillation (SiD) and Diffusion GAN. SiDA accelerates the generation process, significantly reducing the number of iterations required compared to conventional score distillation methods, while producing high-quality images. Extensive tests across various datasets—CIFAR-10, ImageNet (64x64 and 512x512), FFHQ, AFHQ-v2—and different scales of pretrained EDM2 models demonstrate that SiDA consistently surpasses both the base teacher diffusion models and other distilled variants. This is validated by superior Fréchet Inception Distance scores and other performance metrics. Notably, SiDA achieves these results without utilizing classifier-free guidance (Ho & Salimans, 2022), suggesting potential for further enhancement with its integration. Specifically, incorporating the long-and-short guidance strategy as described by Zhou et al. (2024a) could further refine SiDA's performance. Future work will explore expanding SiDA's methodology to additional generative tasks and refining its capabilities through innovative distillation techniques.

---

[1] https://github.com/Alpha-VLLM/LLaMA2-Accessory/blob/main/Large-DiT-ImageNet/assets/table.png

[2] MAR also requires 256 auto-regressive steps before performing reverse diffusion

ACKNOWLEDGMENTS

M. Zhou, H. Zheng, Y. Gu, and Z. Wang acknowledge the support of NSF-IIS 2212418 and NIH-R37 CA271186.

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

# Appendix for SiDA

## A  EXPERIMETAL SETTINGS

**Implementation Details.**    We implement SiDA using the SiD codebase (Zhou et al., 2024b), incorporating essential functions adapted from both EDM (Karras et al., 2022) and EDM2 (Karras et al., 2024). Detailed adaptations of EDM2 modifications within SiD and SiDA are provided in Appendix B.2, specifically addressing our handling of forced weight normalization and data-specific weighting for the loss terms introduced by EDM2 to enhance EDM. We did not incorporate classifier-free guidance (Ho & Salimans, 2022), use two teacher models, or set exponential moving average (EMA) parameters post-hoc, all of which are important for EDM2's performance improvements over EDM. Even without these techniques, SiD of Zhou et al. (2024b) already performs on par with EDM2 teacher models, while SiDA and SiD$^2$A significantly outperform the teacher. These additional techniques could potentially further enhance SiD and SiDA, which we leave for future work.

The score estimation network, $f_\psi$, is initialized using the same architecture and parameters as the pre-trained teacher score network, $f_\phi$, from either EDM or EDM2. For the initialization of the student generator, $G_\theta$, we explore two settings: one where $G_\theta$ is initialized with the same parameters as the pre-trained EDM or EDM2 teacher score network, and another where it is initialized using the distilled generator from SiD, referred to as SiD-SiDA (SiD$^2$A). The hyperparameters customized for our study are outlined in Table 7, with all remaining settings consistent with those in the EDM code (Karras et al., 2022), EDM2 code (Karras et al., 2024), and SiD code (Zhou et al., 2024b).

Our training process unfolds in three stages: 1) For the first 100k images, we exclusively train $f_\psi$ to stabilize the score estimation. 2) For the subsequent 100k images, both $f_\psi$ and $G_\theta$ are trained using the SiD generator loss (6), integrating the distilled insights. 3) For all remaining images, we update $f_\psi$ and $G_\theta$ with the SiDA generator loss (8), which incorporates Diffusion GAN-based adversarial adjustments to refine the generation process. Throughout the training, the score estimation network $f_\psi$ is consistently trained with the SiDA fake score loss (9), ensuring that the adversarial components are properly integrated.

We observe that using only the adversarial loss, without the SiD component in the combined loss, leads to rapid divergence, even when the single-step generator is initialized from SiD. This observation suggests that while the added adversarial components cannot stand alone, they effectively complement the SiD loss, which operates in a data-free manner, allowing it to leverage available training data to correct the teacher's bias and thus enhance the distillation speed and performance.

**Parameter Settings.**    To balance the SiD and adversarial components in both the generator and fake-score losses, we set $\lambda_{\text{sid}}, \lambda_{\text{adv},\theta}, \lambda_{\text{adv},\psi}$ to align the scales of the four different losses. This keeps them approximately within the range of 1 to 10,000 to prevent fp16 overflow/underflow. The settings in Tables 6 and 7 achieve this. For EDM distillation, we reused the SiD parameters. For EDM2, we matched the batch size to the teacher's and adapted $\lambda_{\text{adv},\psi}$ from 1 to 100 to roughly equate the adversarial loss scale with the diffusion distillation loss for the fake score network. Initially, we used a learning rate of 4e-6 but switched to 5e-5 for faster convergence in EDM2-XS. Our experience indicates that SiD and SiDA are not overly sensitive to these settings as long as loss scales are comparable, suggesting robustness. While fine-tuning these hyper-parameters could potentially further improve outcomes, our focus wasn't on hyperparameter optimization given the already achieved state-of-the-art performance across various settings.

**Datasets and Teacher Models.**    To comprehensively evaluate the effectiveness of SiDA, we first utilize four representative datasets of varying scales and resolutions and the EDM diffusion models pretrained on them, as discussed in Karras et al. (2022). These datasets include CIFAR-10 ($32 \times 32$; both conditional and unconditional) (Krizhevsky et al., 2009), ImageNet 64x64 (Deng et al., 2009), FFHQ ($64 \times 64$) (Karras et al., 2019), and AFHQ-v2 ($64 \times 64$) (Choi et al., 2020).

Additionally, we distill EDM2 diffusion models of varying sizes that were pretrained on ImageNet 512x512, following details provided by Karras et al. (2024). We compare these distilled models to their corresponding teacher models and to a concurrent work of Lu & Song (2024) on EDM2 distillation. The models range from 125M million to 1.5 billion parameters. By utilizing these diverse datasets and model scales, we thoroughly assess the performance of SiD, SiDA, and SiD$^2$A across different content types and complexities, ensuring a robust evaluation of their generative capabilities.

Table 6: Hyperparameter settings of SiDA and comparison of distillation time, and memory usage between SiD and SiDA, using 8 NVIDIA V100 GPUs (16 GB each) or 8 NVIDIA H100 GPUs (80 GB each).

| Method | Hyperparameters | CIFAR-10 32x32 | ImageNet 64x64 | FFHQ 64x64 | AFHQ-v2 64x64 |
|---|---|---|---|---|---|
| | Batch size | 256 | 8192 | 512 | 512 |
| | Batch size per GPU | 32 | 32 | 64 | 64 |
| | GPUs | 8xV100 | 8xH100 | 8xH100 | 8xH100 |
| | Gradient accumulation round | 1 | 32 | 1 | 1 |
| | Learning rate of $(\psi, \theta)$ | 1e-5 | 4e-6 | 1e-5 | 5e-6 |
| | Loss scaling of $(\lambda_{\text{sid}}, \lambda_{\text{adv},\theta}, \lambda_{\text{adv},\psi})$ | | | (100, 0.01, 1) | |
| | ema | 0.5 | 2 | 0.5 | 0.5 |
| | fp16 | False | True | True | True |
| | Optimizer Adam (eps) | 1e-8 | 1e-6 | 1e-6 | 1e-6 |
| | Optimizer Adam $(\beta_1)$ | | | 0 | |
| | Optimizer Adam $(\beta_2)$ | | | 0.999 | |
| | $\alpha$ | | 1.0 for SiDA; both 1.0 and 1.2 for SiD$^2$A | | |
| | $\sigma_{\text{init}}$ | | | 2.5 | |
| | $t_{\max}$ | | | 800 | |
| | augment, dropout, cres | | The same as in EDM and SiD for each corresponding dataset | | |
| SiD | max memory allocated per GPU | 13.0 | 46.7 | 31.1 | 31.1 |
| | max memory in GB reserved per GPU | 13.3 | 48.0 | 31.3 | 31.3 |
| | ~seconds per 1k images | 3.3 | 3.1 | 1.1 | 1.1 |
| | ~hours per 10M ($10^4$k) images | 9.2 | 8.6 | 3.1 | 3.1 |
| | ~days per 100M ($10^5$k) images | 3.8 | 3.6 | 1.3 | 1.3 |
| SiDA | max memory allocated per GPU | 13.0 | 46.7 | 31.1 | 31.1 |
| | max memory in GB reserved per GPU | 13.4 | 48.1 | 31.3 | 31.3 |
| | ~seconds per 1k images | 3.6 | 3.5 | 1.2 | 1.2 |
| | ~hours per 10M ($10^4$k) images | 10.0 | 9.7 | 3.3 | 3.3 |
| | ~days per 100M ($10^5$k) images | 4.2 | 4.1 | 1.4 | 1.4 |

Table 7: Hyperparameter settings and comparison of distillation time, and memory usage between SiD and SiDA on distilling EDM2 pretrained on ImageNet 512x512, using 16 NVIDIA A100 GPUs (40 GB each) or 8 NVIDIA H100 GPUs (80 GB each).

| Method | Hyperparameters | EDM2-XS | EDM2-S | EDM2-M | EDM2-L | EDM2-XL | EDM2-XXL |
|---|---|---|---|---|---|---|---|
| | Batch size | | | 2048 | | | |
| | Batch size per GPU | 64 | 64 | 32 | 32 | 16 | 2 |
| | GPUs | 16xA100 | 8xH100 | 8xH100 | 8xH100 | 8xH100 | 8xH100 |
| | Gradient accumulation round | 2 | 4 | 8 | 8 | 16 | 128 |
| | Learning rate of $(\psi, \theta)$ | | | 5e-5 | | | |
| | Loss scaling of $(\lambda_{\text{sid}}, \lambda_{\text{adv},\theta}, \lambda_{\text{adv},\psi})$ | | | (100, 0.01, 100) | | | |
| | ema | | | 2 | | | |
| | fp16 | | | True | | | |
| | Optimizer Adam $(\beta_1, \beta_2, \text{eps})$ | | | (0, 0.999, 1e-6) | | | |
| | $\alpha$ | | | 1.0 | | | |
| | $\sigma_{\text{init}}$ | | | 2.5 | | | |
| | $t_{\max}$ | | | 800 | | | |
| | dropout | | The same as in EDM2 for each corresponding model size | | | | |
| SiD | max memory allocated per GPU | 31.5 | 51.3 | 49.4 | 68.8 | 72.2 | 74.6 |
| | max memory in GB reserved per GPU | 32.3 | 52.4 | 50.0 | 70.0 | 73.4 | 75.1 |
| | ~seconds per 1k images | 0.91 | 1.5 | 2.5 | 3.4 | 5.7 | 29.9 |
| | ~hours per 10M ($10^4$k) images | 2.5 | 4.2 | 6.9 | 9.4 | 15.8 | 83.1 |
| | ~days per 100M ($10^5$k) images | 1.1 | 1.7 | 2.9 | 3.9 | 6.6 | 35 |
| SiDA | max memory allocated per GPU | 31.5 | 51.3 | 49.4 | 68.9 | 72.2 | 74.6 |
| | max memory in GB reserved per GPU | 34.2 | 52.4 | 52.0 | 70.1 | 73.4 | 75.1 |
| | ~seconds per 1k images | 1.0 | 1.6 | 2.7 | 3.7 | 6.1 | 31.2 |
| | ~hours per 10M ($10^4$k) images | 2.7 | 4.4 | 7.5 | 10.3 | 16.9 | 86.7 |
| | ~days per 100M ($10^5$k) images | 1.2 | 1.9 | 3.1 | 4.3 | 7.1 | 36.1 |

**Evaluation Protocol.** We assess the quality of image generation using the Fréchet Inception Distance (FID) and Inception Score (IS) (Salimans et al., 2016). Following Karras et al. (2024), we also evaluate models trained on ImageNet 512x512 using $FD_{\text{DINOv2}}$, the Fréchet distance computed in the feature space of DINOv2 (Oquab et al., 2023). Consistent with the methodology of Karras et al. (2019; 2022; 2024), these scores are calculated using 50,000 generated samples, with the training dataset used by the EDM or EDM2 teacher model[3] serving as the reference. Additionally, we evaluate SiDA on ImageNet 64x64 using the Precision and Recall metrics (Kynkäänniemi et al., 2019), where

---

[3]https://github.com/NVlabs/edm

both metrics are calculated using a predefined reference batch[4][5] (Dhariwal & Nichol, 2021a; Nichol & Dhariwal, 2021; Song et al., 2023; Song & Dhariwal, 2023).

In line with SiD, we periodically evaluate the FID during distillation and retain the generators with the lowest FID. To ensure accuracy and statistically robust comparisons, we perform re-evaluations across 10 independent runs to obtain more reliable metrics. We recommend this rigorous approach over reporting only the best metrics observed across individual runs or during the distillation process, as the latter can result in biased metrics that are consistently lower than the mean.

# B  ALGORITHM DETAILS

## B.1  ALGORITHM BOX

---

**Algorithm 1** Adversarial Score identity Distillation (SiDA)

---

1: **Input:** Pretrained score network $f_\phi$, generator $G_\theta$, generator score network $f_\psi$, $\sigma_{\text{init}} = 2.5$, $t_{\max} = 800$, $\alpha = 1.2$, $\lambda_{\text{sid}} = 100$, $\lambda_{\text{adv},\theta} = 0.01$, $\lambda_{\text{adv},\psi} = 1$, image size $(W, H, C_c)$, latent discriminator map size $(W', H')$.

2: **Initialization** $\theta \leftarrow \phi, \psi \leftarrow \phi, D(\cdot) \leftarrow \text{encoder}(\psi)$

3: **repeat**

4:     Sample $\boldsymbol{z} \sim \mathcal{N}(0, \mathbf{I})$ and $\boldsymbol{x}_g = G_\theta(\sigma_{\text{init}}\boldsymbol{z})$

5:     Sample $t \sim p(t)$, $\boldsymbol{\epsilon}_t \sim \mathcal{N}(0, \mathbf{I})$ and $\boldsymbol{x}_t = a_t \boldsymbol{x}_g + \sigma_t \boldsymbol{\epsilon}_t$

6:     Update $\psi$ with 9:

7:         $\mathcal{L}_\psi^{(\text{sida})} = \gamma(t)(\|f_\psi(\boldsymbol{x}_t, t) - \boldsymbol{x}_g\|_2^2 + \lambda_{\text{adv},\psi} L_\psi^{(\text{adv})})$

8:         $\psi = \psi - \eta \nabla_\psi \mathcal{L}_\psi^{(\text{sida})}$

9:     where the timestep distribution $t \sim p(t)$, noise level $\sigma_t$, and weighting function $\gamma(t)$ are defined as in Zhou et al. (2024b).

10:     **if** num_imgs $\geq$ 100K **then**

11:         Sample $t \sim \text{Unif}[0, t_{\max}/1000]$, compute $\sigma_t$, $\omega_t$, and $a_t$ as defined in Zhou et al. (2024b)

12:         Set $b = 0$ if num_imgs $\leq$ 200k and $b = 1$ otherwise

13:         Update $G_\theta$ with 8:

14:         $\mathcal{L}_\theta^{(\text{sida})} = \left(\frac{1}{2}\right)^b \lambda_{\text{sid}} \left( (1 - \alpha)\frac{\omega(t)a_t^2}{\sigma_t^4}\|f_\phi(\boldsymbol{x}_t, t) - f_\psi(\boldsymbol{x}_t, t)\|^2 \right.$

15:         $\left. + \frac{\omega(t)a_t^2}{\sigma_t^4}(f_\phi(\boldsymbol{x}_t, t) - f_\psi(\boldsymbol{x}_t, t))^T(f_\psi(\boldsymbol{x}_t, t) - \boldsymbol{x}_g) \right) + \frac{b}{2}\lambda_{\text{adv},\theta}\frac{\omega(t)a_t^2}{2\sigma_t^4}C_c W H \mathcal{L}_\theta^{(\text{adv})}$

16:         $\theta = \theta - \eta \nabla_\theta \mathcal{L}_\theta^{(\text{sida})}$

17:     **end if**

18: **until** the FID plateaus or the training budget is exhausted

19: **Output:** $G_\theta$

---

---

**Algorithm 2** SiDA for EDM2

---

1: **Extra Input:** $\text{LogVar}_{\text{Flag}} \in \{\text{True}, \text{False}\}$, $\text{ForceNorm}_{\text{Flag}} \in \{\text{True}, \text{False}\}$, $\lambda_{\text{adv},\psi} = 100$

2: **repeat**

3:     The same as SiDA for EDM with the following modifications:

4:     **if** $\text{LogVar}_{\text{Flag}}$ is True **then**

5:         Modify the fake score network loss as $\mathcal{L}_\psi^{(\text{sida})} = \frac{\gamma(t)}{e^{\text{logvar}}}(\|f_\psi(\boldsymbol{x}_t, t) - \boldsymbol{x}_g\|_2^2 + \text{logvar} + \lambda_{\text{adv},\psi} L_\psi^{(\text{adv})})$

6:     **end if**

7:     **if** $\text{ForceNorm}_{\text{Flag}}$ is True **then**

8:         During training, apply forced weight normalization used by EDM2 for SiD, and a modified version of it to prevent gradient backpropagation error for SiDA

9:     **end if**

10: **until** the FID plateaus or the training budget is exhausted

11: **Output:** $G_\theta$

---

---

[4]https://openaipublic.blob.core.windows.net/diffusion/jul-2021/ref_batches/imagenet/64/VIRTUAL_imagenet64_labeled.npz

[5]https://github.com/openai/guided-diffusion/tree/main/evaluations

---

**Algorithm 3** SiD$^2$A: SiD initilized SiDA (applicable for both EDM and EDM2)

---

1: **Extra Input:** Pretrained SiD generator $\theta_{\text{sid}}$
2: **Initialization** $\theta \leftarrow \theta_{\text{sid}}, \psi \leftarrow \phi$ , $D(\cdot) \leftarrow \text{encoder}(\psi)$
3: **repeat**
4:    The same as SiDA
5: **until** the FID plateaus or the training budget is exhausted
6: **Output:** $G_\theta$

---

### B.2 Forced Weight Normalization with Pre-Hook

An important training technique used by EDM2 to enhance EDM is the application of *forced weight normalization*, where each weight vector is explicitly normalized to unit variance before every training step. This method is found to be fully compatible with the training of both the generator and fake score network in SiD and does not introduce gradient backpropagation errors. However, in SiDA, where adversarial loss is incorporated, we find that this approach often leads to computational errors during gradient backpropagation.

We have traced the source of the problem to the `MPConv` class of the EDM2 code, which modifies `self.weight` in-place during the forward pass. A possible reason is that even though `torch.no_grad()` was used, in-place operations on parameters that require gradients can interfere with PyTorch's autograd system, leading to errors during backpropagation.

To address these issues, we provide two options:

1. **Disable Forced Weight Normalization**: Completely remove the in-place normalization step from the training process. This avoids the gradient backpropagation errors but also forgoes the potential benefits of forced weight normalization.

2. **Use a Pre-Forward Hook to Avoid In-Place Operations**: Instead of modifying `self.weight` in-place during the forward pass, we utilize a pre-forward hook to normalize the weights before each forward pass. Executed before the forward method of a module, a pre-hook in PyTorch allows for consistent normalization of the weights without interfering with the autograd system. Employing this strategy enables forced weight normalization without causing gradient backpropagation problems in SiDA.

By implementing the second option, we maintain the potential advantages of forced weight normalization while ensuring stable training with adversarial loss in SiDA.

In our experiments, we assessed two options during the distillation of EDM2-XS, EDM2-S, and EDM2-M and observed no significant performance differences. For SiD, we retained the original forced weight normalization. For SiDA, we generally opted for forced weight normalization with pre-hook (option 2), except for EDM2-S and EDM2-XXL where we used no forced weight normalization (option 1). Similarly, SiD$^2$A defaults to option 2, but switches to option 1 for EDM2-S. These variations in choices are not intended to maximize performance for each setting, but rather to illustrate that the choice of normalization may not significantly impact results, or at least not impede their ability to outperform the teacher.

Another clear difference between EDM2 and EDM is the use of a one-layer MLP to produce the weights for each data-specific loss term. This MLP is trained alongside the main denoising network and discarded afterward. In our experiments, we did not observe a clear difference in distillation performance with or without this MLP. Therefore, we enable it by default when forced weight normalization is applied and disable it when no forced normalization is used.

Below, we provide the code for the original and modified versions of the `MPConv` class.

**The Original `MPConv` Class of EDM2 That Includes Forced Weight Normalization**

```
@persistence.persistent_class
class MPConv(torch.nn.Module):
    def __init__(self, in_channels, out_channels, kernel):
        super().__init__()
        self.out_channels = out_channels
        self.weight = torch.nn.Parameter(torch.randn(out_channels, in_channels, *kernel))

    def forward(self, x, gain=1):
        w = self.weight.to(torch.float32)
        if self.training:
            with torch.no_grad():
                self.weight.copy_(normalize(w)) # forced weight normalization
        w = normalize(w) # traditional weight normalization
        w = w * (gain / np.sqrt(w[0].numel())) # magnitude-preserving scaling
        w = w.to(x.dtype)
        if w.ndim == 2:
            return x @ w.t()
        assert w.ndim == 4
        return torch.nn.functional.conv2d(x, w, padding=(w.shape[-1]//2,))
```

Listing 1: Original `MPConv` class with forced weight normalization

**The Modified `MPConv` Class Introducing an Optional Modified Forced Weight Normalization When Adversarial Loss Is Applied**

```
@persistence.persistent_class
class MPConv(torch.nn.Module):
    def __init__(self, in_channels, out_channels, kernel,force_normalization,use_gan):
        super().__init__()
        self.out_channels = out_channels
        self.weight = torch.nn.Parameter(torch.randn(out_channels, in_channels, *kernel))
        self.force_normalization =force_normalization
        self.use_gan = use_gan
        # Register the forward pre-hook
        if self.use_gan and self.force_normalization:
            self.register_forward_pre_hook(self._apply_forced_weight_normalization)

    def _apply_forced_weight_normalization(self, module, input):
        # Only apply during training
        if self.training:
            with torch.no_grad():
                w = self.weight.to(torch.float32)
                w_normalized = normalize(w)
                self.weight.copy_(w_normalized)

    def forward(self, x, gain=1):
        w = self.weight.to(torch.float32)
        if self.training and not self.use_gan:
            with torch.no_grad():
                self.weight.copy_(normalize(w)) # forced weight normalization
        w = normalize(w)  # Traditional weight normalization
        w = w * (gain / np.sqrt(w[0].numel()))  # Magnitude-preserving scaling
        w = w.to(x.dtype)
        if w.ndim == 2:
            return x @ w.t()
        assert w.ndim == 4
        return torch.nn.functional.conv2d(x, w, padding=(w.shape[-1] // 2))
```

Listing 2: Modified `MPConv` class with pre-forward hook

# C    ADDITIONAL TRAINING AND EVALUATION DETAILS

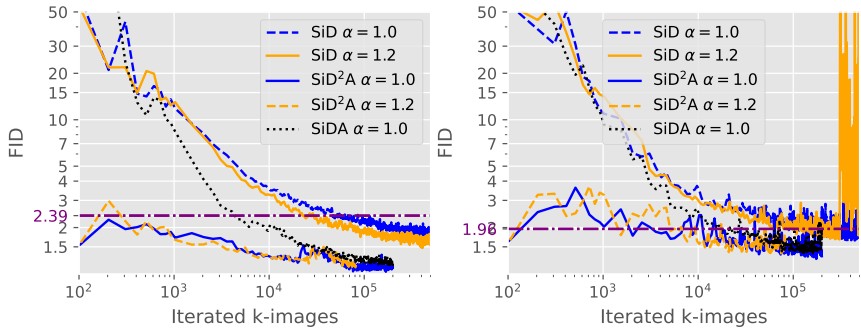

Figure 6: Analogous plot to Figure 4 for FFHQ-64x64 with batch size of 512 (*left*) and AFHQ-v2-64x64 with batch size of 512 (*right*).

Table 8: Analogous to Table 1 for FFHQ 64x64.

| Family | Model | NFE | FID ($\downarrow$) |
|---|---|---|---|
| Teacher | VP-EDM (Karras et al., 2022) | 79 | 2.39 |
| Diffusion | VP-EDM (Karras et al., 2022) | 50 | 2.60 |
| | Patch-Diffusion (Wang et al., 2023a) | 50 | 3.11 |
| Distillation | BOOT (Gu et al., 2023) | 1 | 9.00 |
| | SiD, $\alpha = 1.0$ | 1 | $1.710 \pm 0.018$ |
| | SiD, $\alpha = 1.2$ | 1 | $1.550 \pm 0.017$ |
| | SiDA, $\alpha = 1.0$ (ours) | 1 | $1.134 \pm 0.012$ |
| | SiD$^2$A, $\alpha = 1.0$ (ours) | 1 | **1.040** $\pm 0.011$ |
| | SiD$^2$A, $\alpha = 1.2$ (ours) | 1 | $1.109 \pm 0.015$ |

Table 9: Analogous to Table 1 for AFHQ-v2 64x64.

| Family | Model | NFE | FID ($\downarrow$) |
|---|---|---|---|
| Teacher | VP-EDM (Karras et al., 2022) | 79 | 1.96 |
| Distillation | SiD, $\alpha = 1.0$ | 1 | $1.628 \pm 0.017$ |
| | SiD, $\alpha = 1.2$ | 1 | $1.711 \pm 0.020$ |
| | SiDA, $\alpha = 1.0$ (ours) | 1 | $1.345 \pm 0.015$ |
| | SiD$^2$A, $\alpha = 1.0$ (ours) | 1 | **1.276** $\pm 0.010$ |
| | SiD$^2$A, $\alpha = 1.2$ (ours) | 1 | $1.366 \pm 0.018$ |

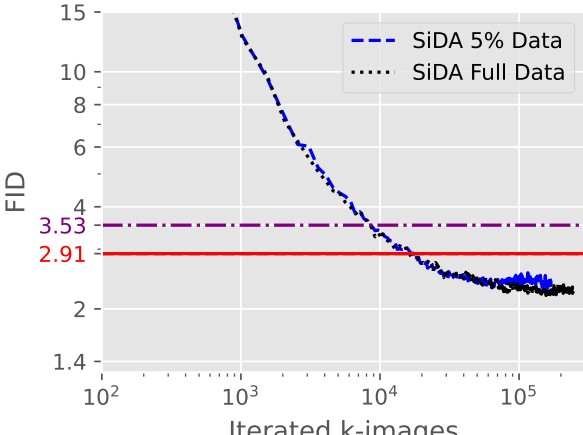

Figure 7: Evolution of FID scores for the SiDA generator during the distillation of the EDM2-XS teacher model, pretrained on ImageNet 512×512. Training utilized either 5% or 100% of the real training images, with a batch size of 2048 and $\alpha = 1.0$. Horizontal lines indicate the performance of EDM2 without classifier-free guidance (CFG) in purple, using 63 NFEs, and with CFG in red, using 63x2 NFEs.

Table 10: FD$_{\text{DINOv2}}$ scores and number of parameters for different methods on ImageNet 512x512.

| Method | CFG | NFE | XS (125M) | S (280M) | M (498M) | L (777M) | XL (1.1B) | XXL (1.5B) |
|---|---|---|---|---|---|---|---|---|
| EDM2 | N | 63 | 103.39 | 68.64 | 58.44 | 52.25 | 45.96 | 42.84 |
| EDM2 | Y | 63x2 | 79.94 | 52.32 | 41.98 | 38.20 | 35.67 | 33.09 |
| SiD | N | 1 | $91.75 \pm 0.55$ | $65.08 \pm 0.32$ | $55.92 \pm 0.25$ | $56.25 \pm 0.40$ | $52.47 \pm 0.39$ | $56.15 \pm 0.41$ |
| SiDA | N | 1 | $93.67 \pm 0.40$ | $68.76 \pm 0.45$ | $53.40 \pm 0.60$ | $48.47 \pm 0.45$ | $44.47 \pm 0.39$ | $50.22 \pm 0.43$ |
| SiD$^2$A | N | 1 | $86.96 \pm 0.32$ | $62.19 \pm 0.36$ | $49.01 \pm 0.54$ | $46.80 \pm 0.29$ | $43.89 \pm 0.51$ | $44.52 \pm 0.37$ |

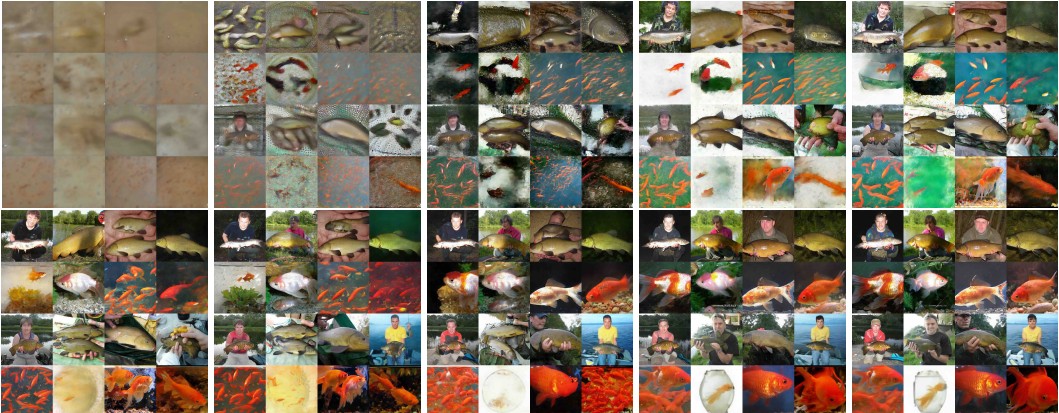

Figure 8: The SiD method highlights rapid advancements in distilling EDM2-XL pretrained on ImageNet 512x512, utilizing a batch size of 2048 and a learning rate of 5e-5. Illustrations are provided using random generations with class labels: 0 for tench and 1 for goldfish. The series of images, generated from the same set of random noises post-training the SiDA generator with varying counts of synthesized images, illustrates progressions at 0, 0.2, 0.5, 1, 2, 4,10, 20, 100, and 200 million images. These are equivalent to 0, 100, 250, 500, 1k, 2K, 5K, 10k, 50k, and 100k training iterations respectively, organized from the top left to the bottom right. The blue doshed curve in Panel (e) of Figure 5 details the progression of FIDs for SiD on EDM2-XL.

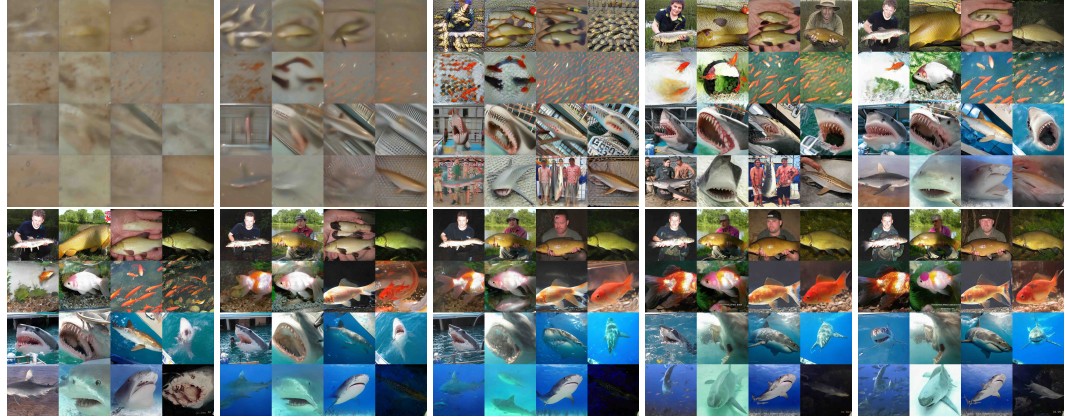

Figure 9: Analogous plot to Figure 8 for the the newly proposed SiDA method, with two more classes introduced for visulization: 2 for great white shark, and 3 for tiger shark. The black dotted curve in Panel (e) of Figure 5 details the progression of FIDs for SiDA on EDM2-XL.

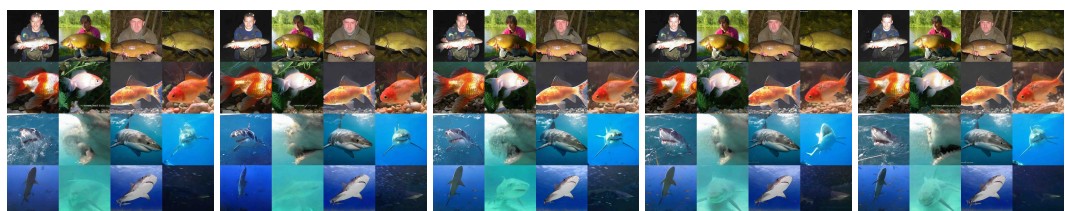

Figure 10: Analogous plot to Figure 10 for the the newly proposed SiD$^2$A method. The series of images, generated from the same set of random noises post-training the SiDA generator with varying counts of synthesized images, illustrates progressions at 0, 4, 10, 20, 40 million images. These are equivalent to 0, 2K, 5K, 10k, 20k training iterations respectively, organized from the left to right. The orange solid curve in Panel (e) of Figure 5 details the progression of FIDs for SiD$^2$A on EDM2-XL.

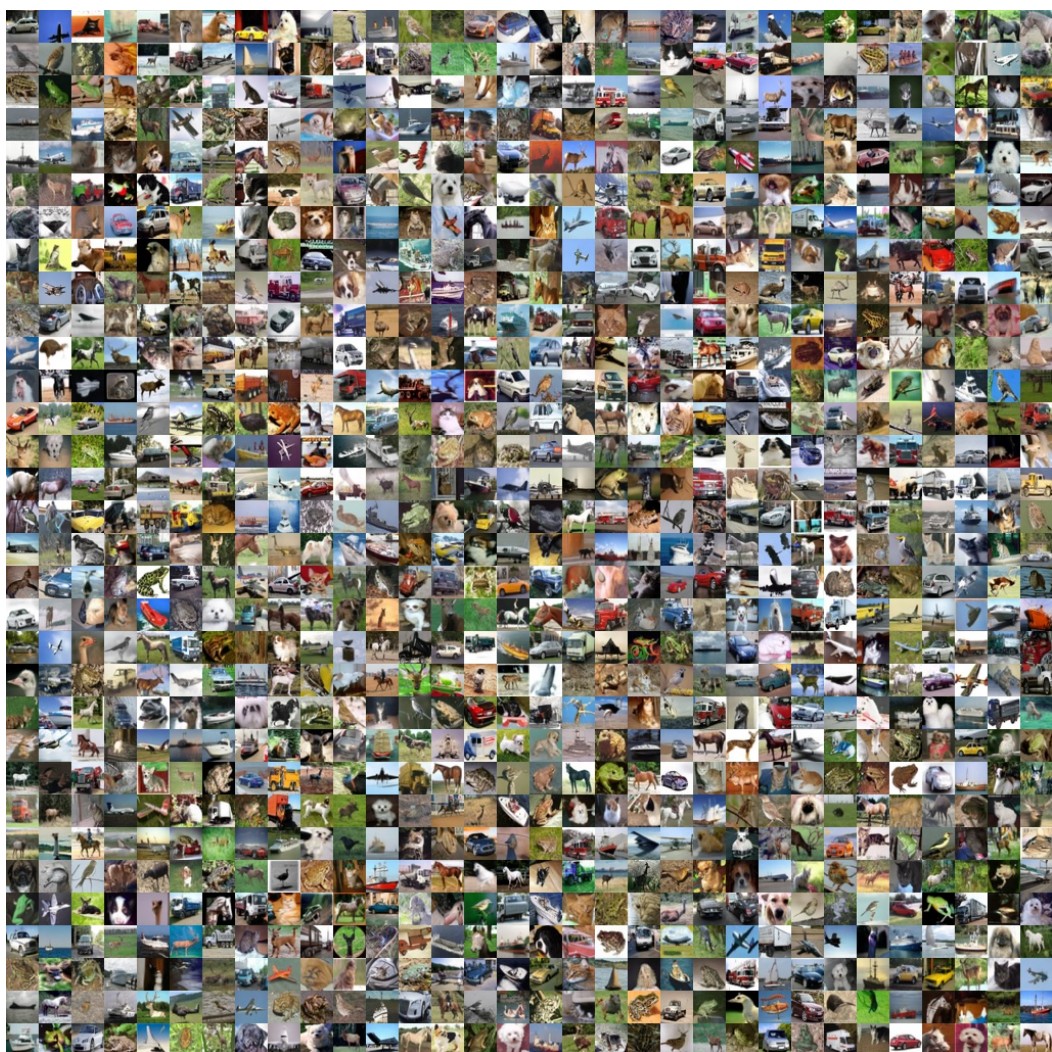

Figure 11: Unconditional CIFAR-10 32X32 random images generated with SiD$^2$A (FID: 1.499).

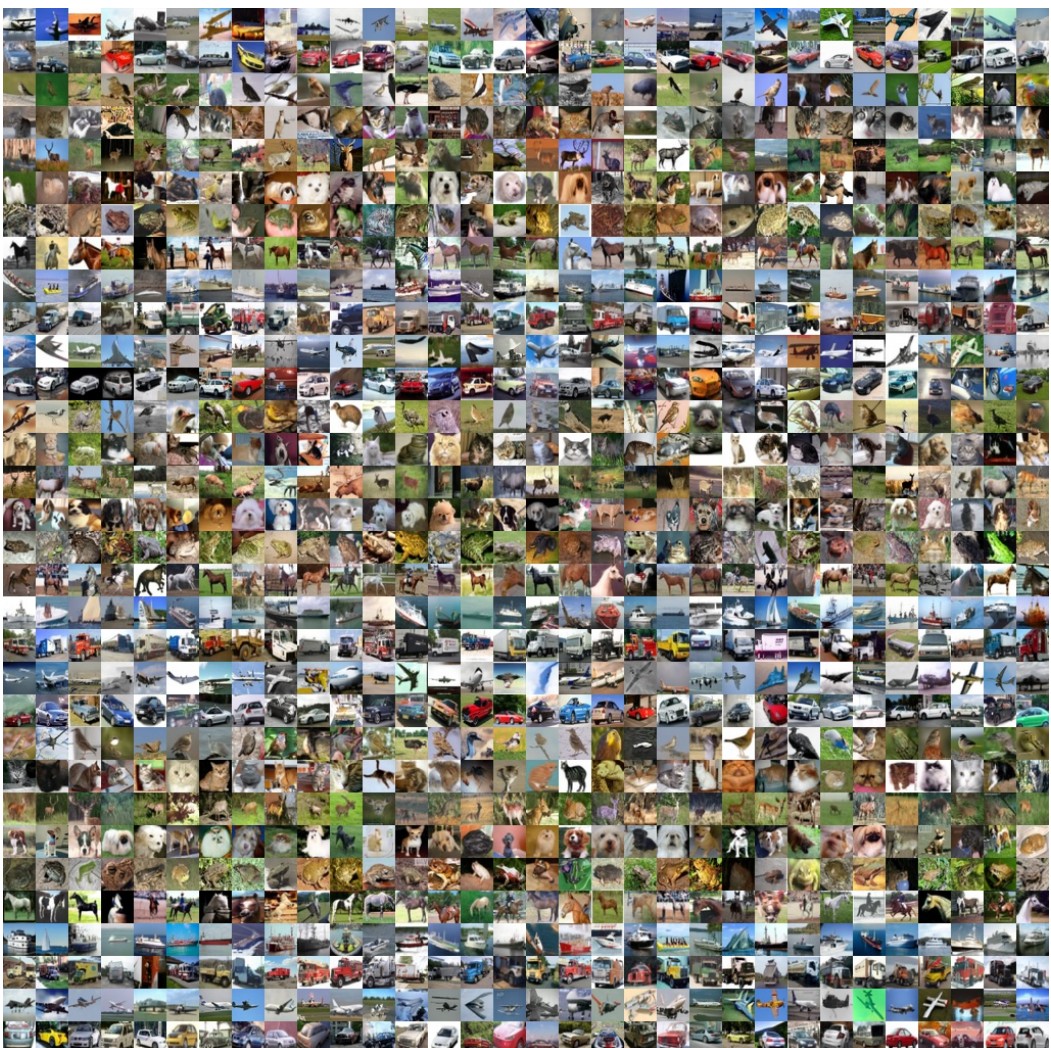

Figure 12: Label conditioning CIFAR-10 32X32 random images generated with SiD$^2$A (FID: 1.396)

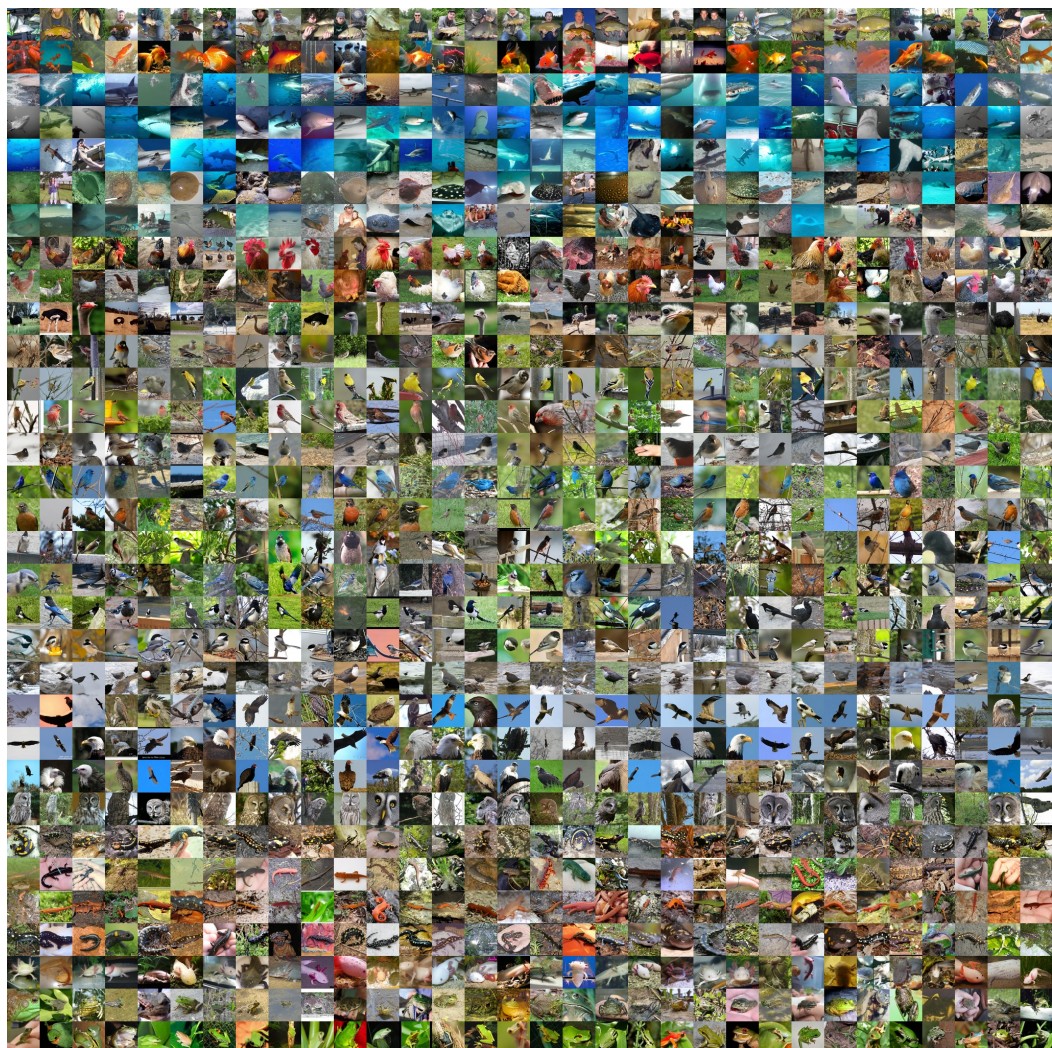

Figure 13: Label conditioning ImageNet 64x64 random images generated with SiD$^2$A (FID: 1.110)

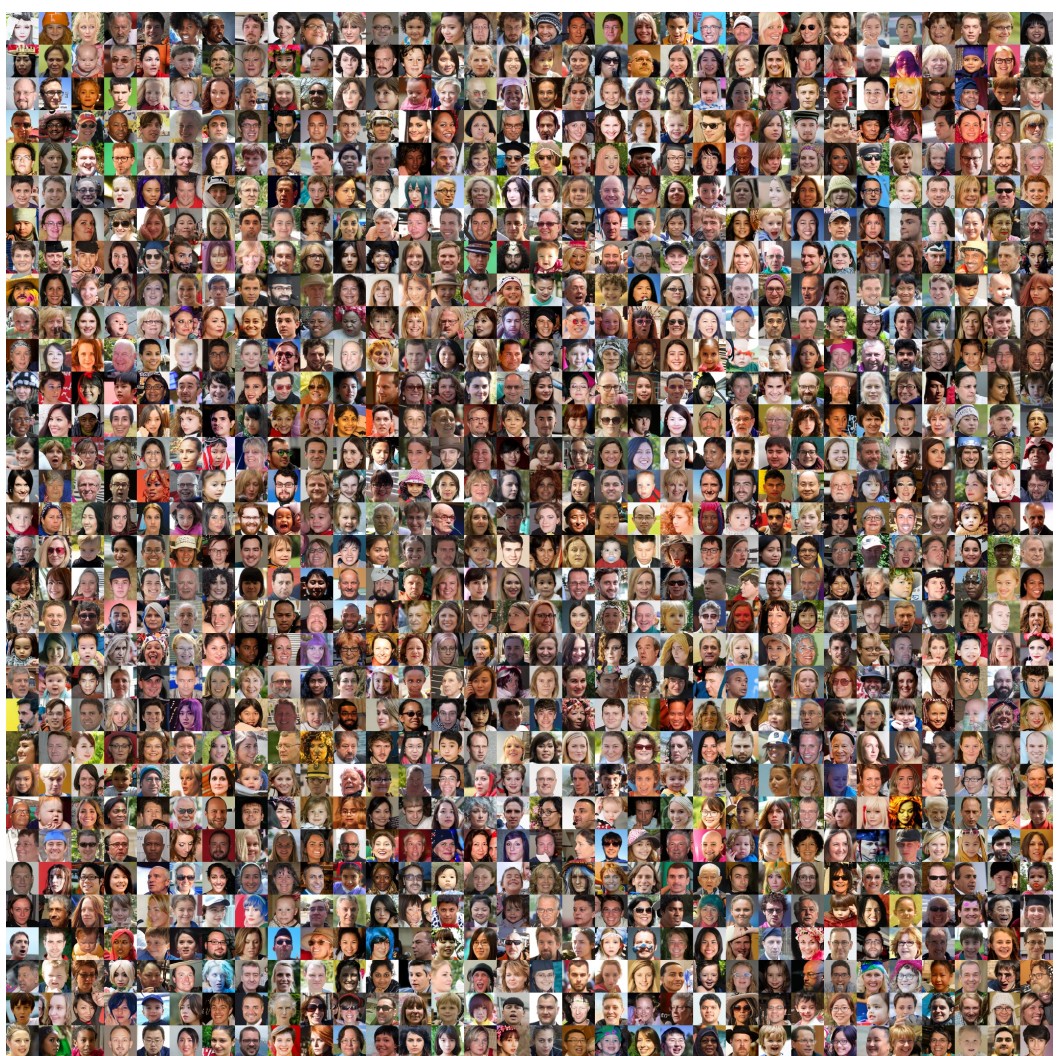

Figure 14: FFHQ 64X64 random images generated with SiD$^2$A (FID: 1.040)

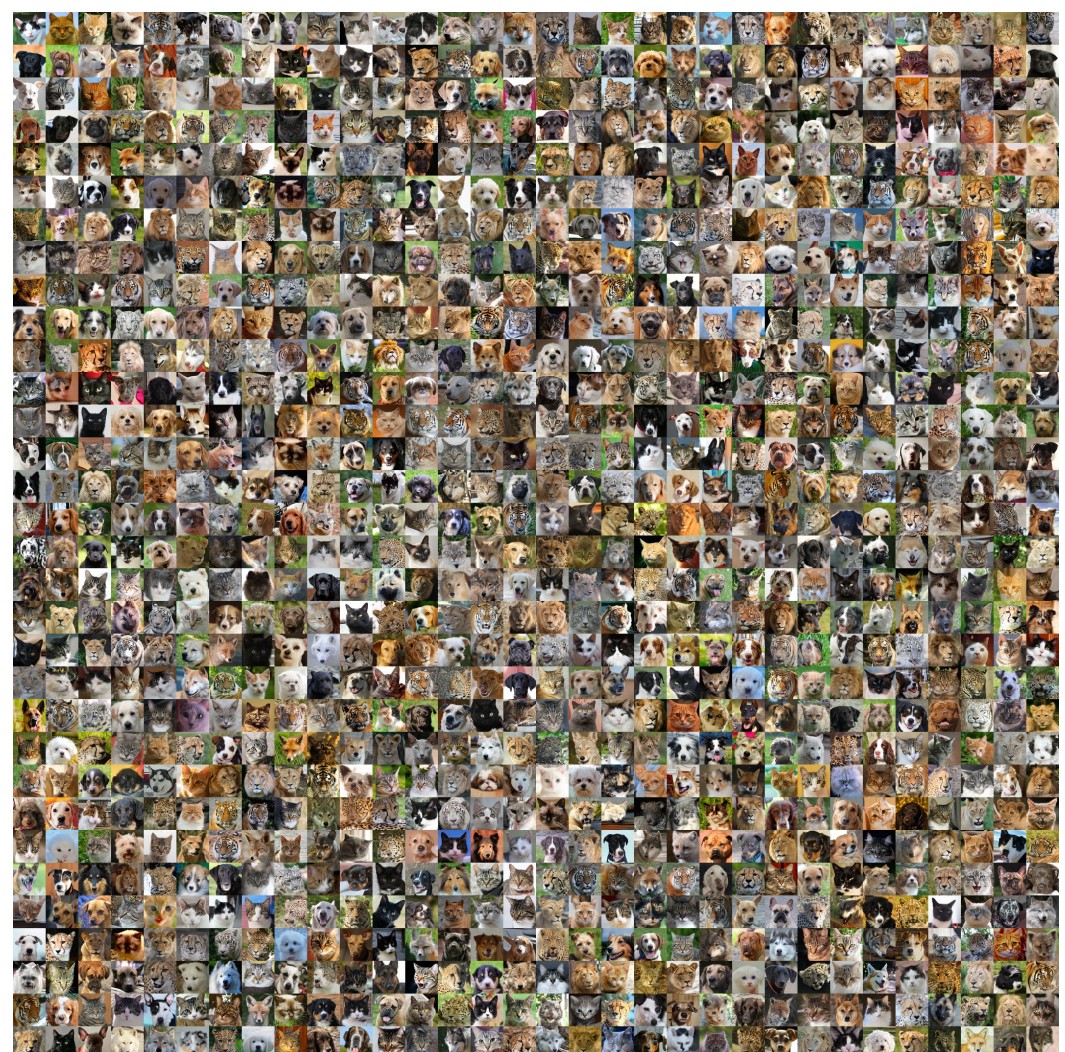

Figure 15: AFHQ-V2 64X64 random images generated with SiD$^2$A (FID: 1.276)

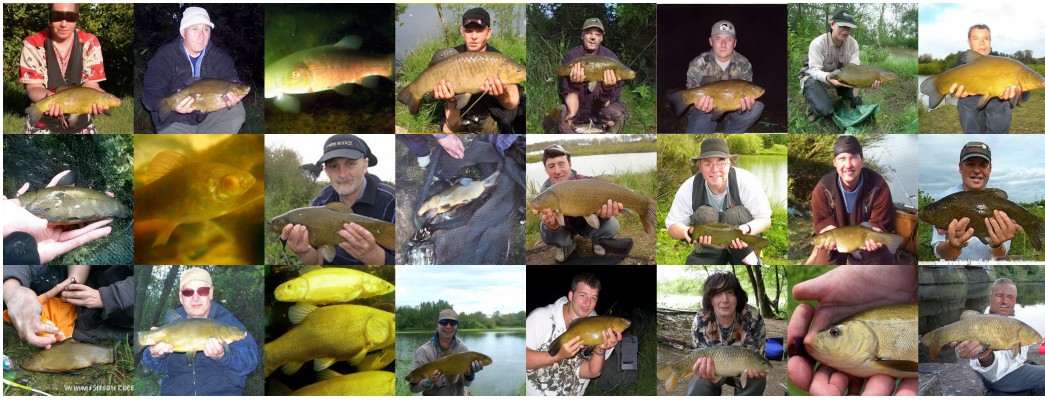

Figure 16: ImageNet 512x512 images of Class 0 (tench) generated using SiD$^2$A without classifier-free guidance, produced in a single generation step.

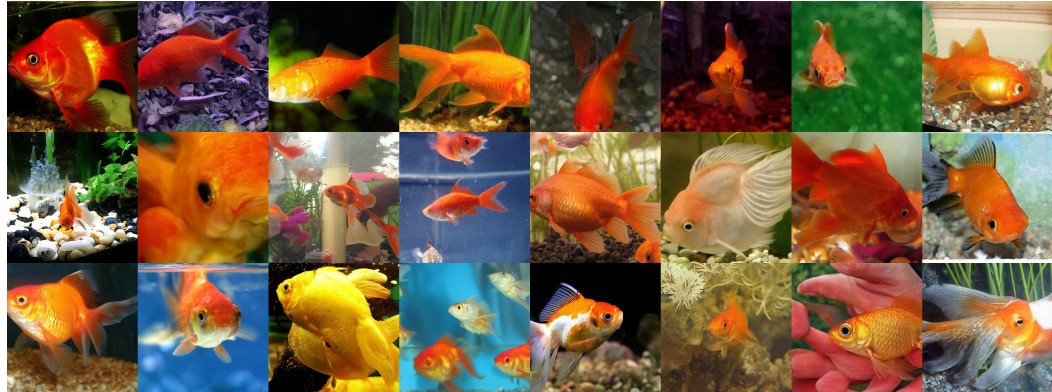

Figure 17: ImageNet 512x512 images of Class 1 (goldfish) generated using SiD$^2$A without classifier-free guidance, produced in a single generation step.

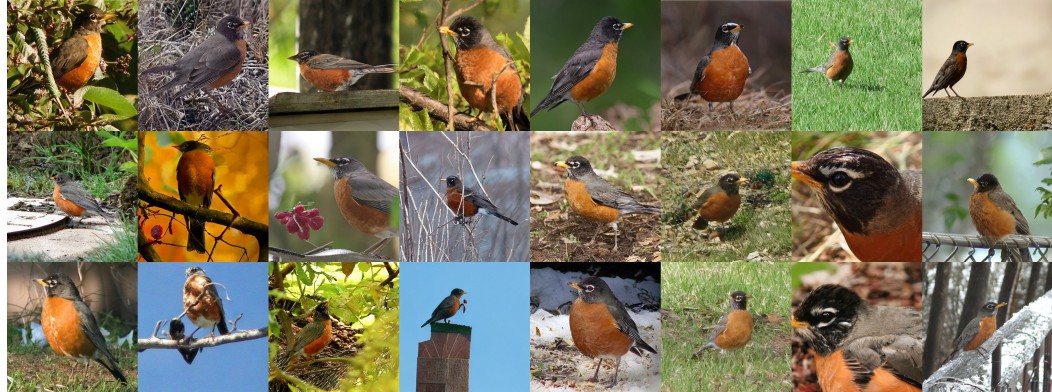

Figure 18: ImageNet 512x512 images of Class 15 (robin) generated using SiD$^2$A without classifier-free guidance, produced in a single generation step.

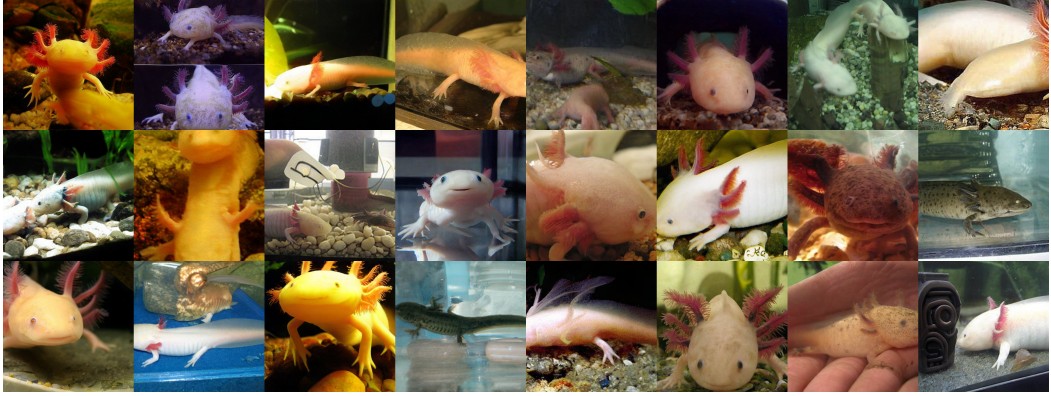

Figure 19: ImageNet 512x512 images of Class 29 (axolotl) generated using SiD$^2$A without classifier-free guidance, produced in a single generation step.

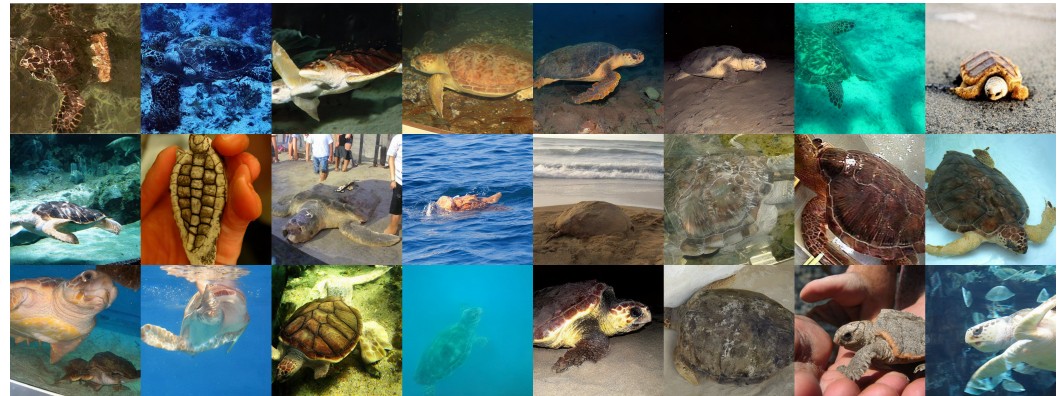

Figure 20: ImageNet 512x512 images of Class 33 (loggerhead) generated using SiD$^2$A without classifier-free guidance, produced in a single generation step.

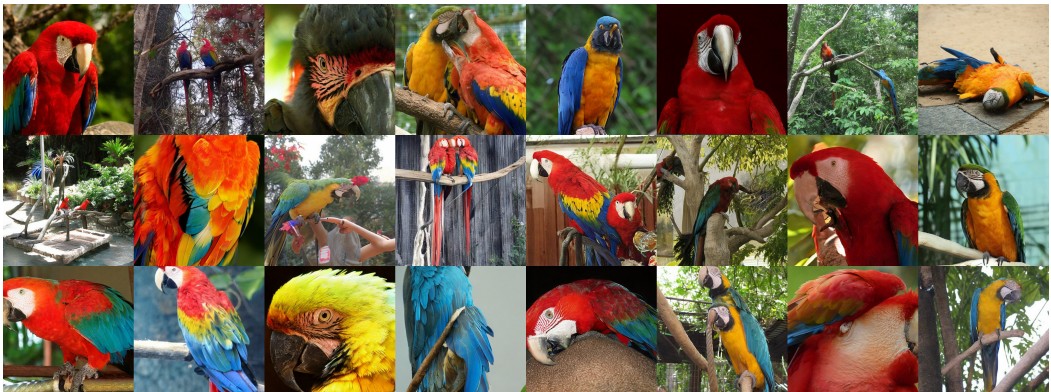

Figure 21: ImageNet 512x512 images of Class 88 (macaw) generated using SiD$^2$A without classifier-free guidance, produced in a single generation step.

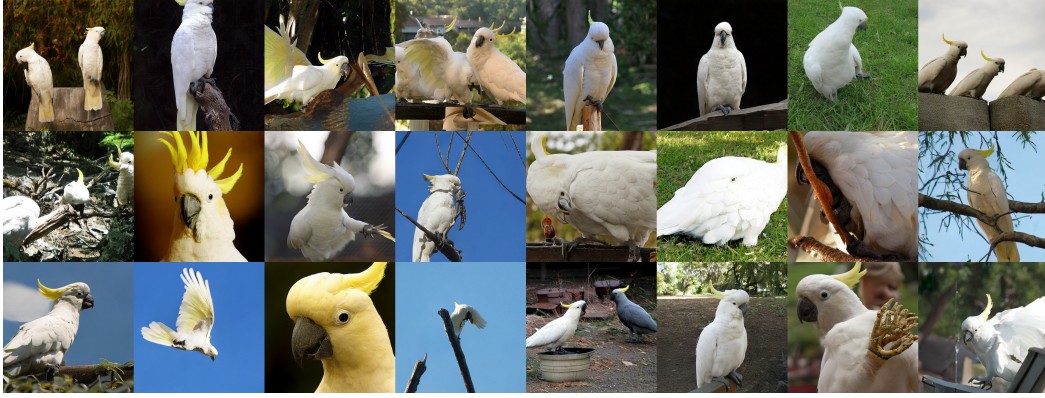

Figure 22: ImageNet 512x512 images of Class 89 (sulphur-crested cockatoo) generated using SiD$^2$A without classifier-free guidance, produced in a single generation step.

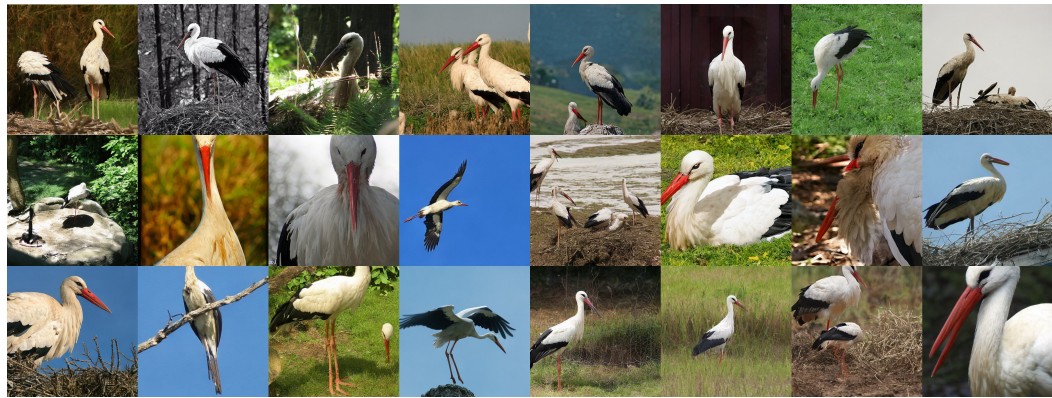

Figure 23: ImageNet 512x512 images of Class 127 (white stork) generated using SiD$^2$A without classifier-free guidance, produced in a single generation step.

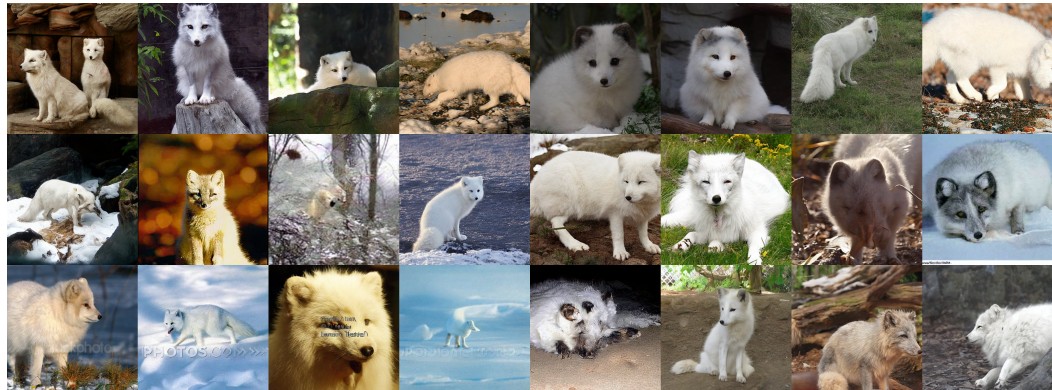

Figure 24: ImageNet 512x512 images of Class 279 (arctic fox) generated using SiD$^2$A without classifier-free guidance, produced in a single generation step.

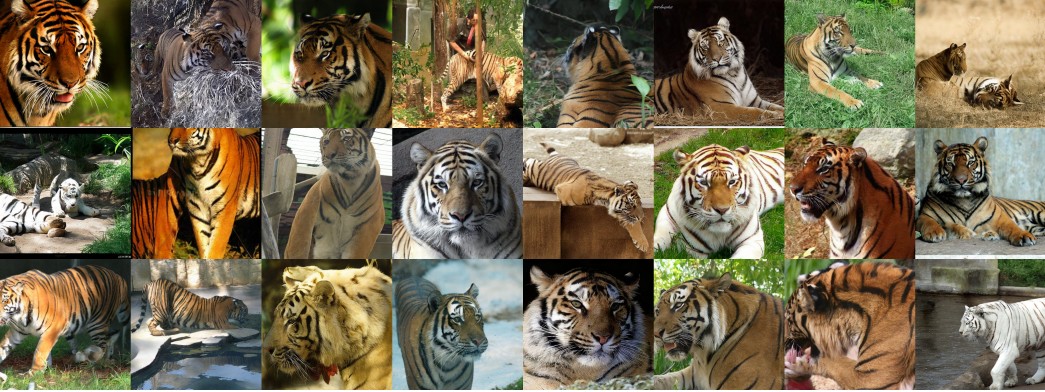

Figure 25: ImageNet 512x512 images of Class 292 (tiger) generated using SiD$^2$A without classifier-free guidance, produced in a single generation step.

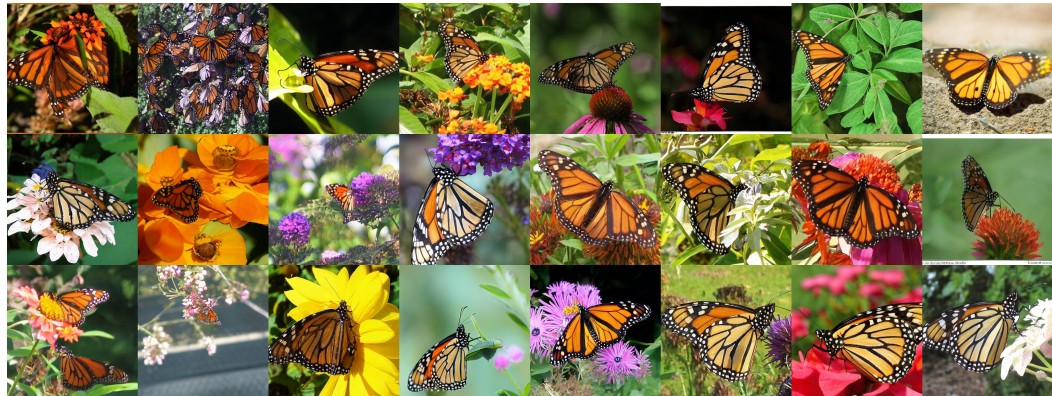

Figure 26: ImageNet 512x512 images of Class 323 (monarch) generated using SiD$^2$A without classifier-free guidance, produced in a single generation step.

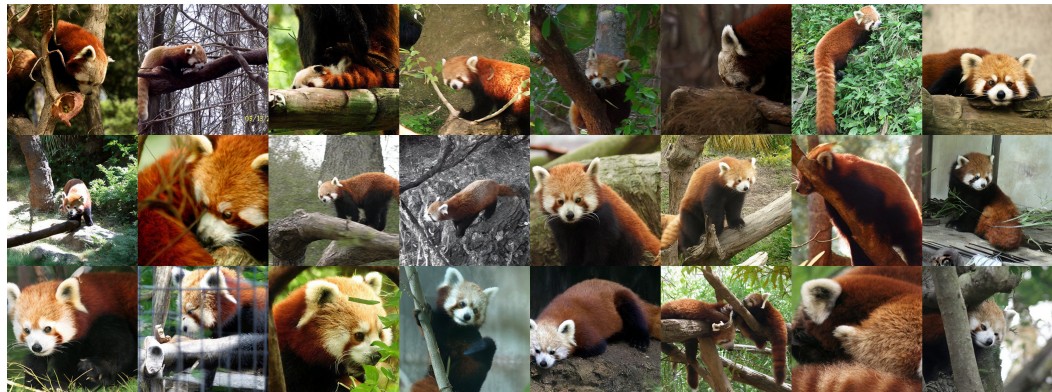

Figure 27: ImageNet 512x512 images of Class 387 (lesser panda) generated using SiD$^2$A without classifier-free guidance, produced in a single generation step.

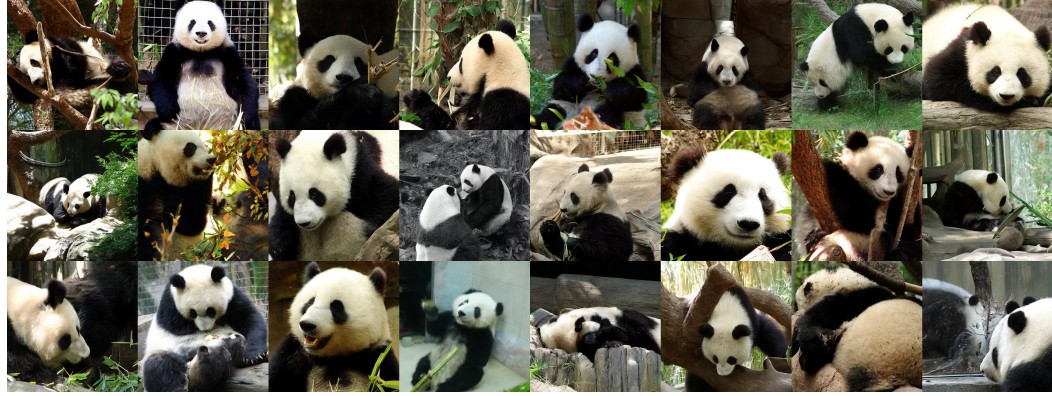

Figure 28: ImageNet 512x512 images of Class 388 (giant panda) generated using SiD$^2$A without classifier-free guidance, produced in a single generation step.

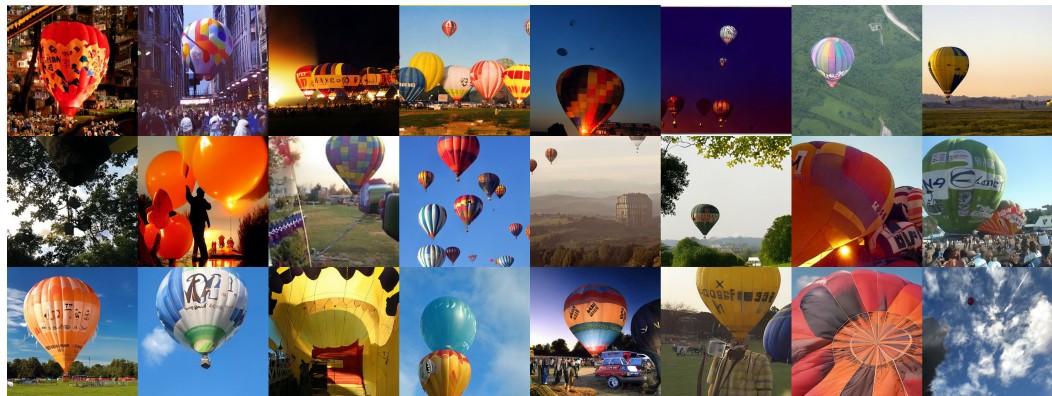

Figure 29: ImageNet 512x512 images of Class 417 (balloon) generated using SiD$^2$A without classifier-free guidance, produced in a single generation step.

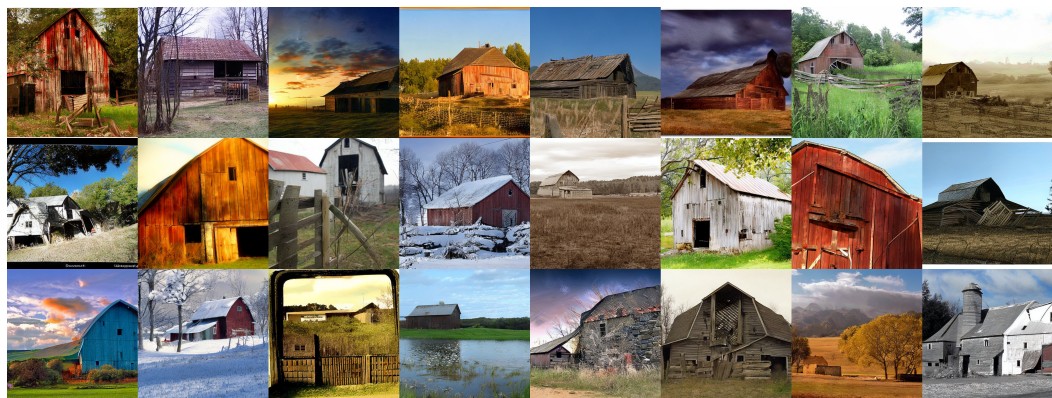

Figure 30: ImageNet 512x512 images of Class 425 (barn) generated using SiD$^2$A without classifier-free guidance, produced in a single generation step.

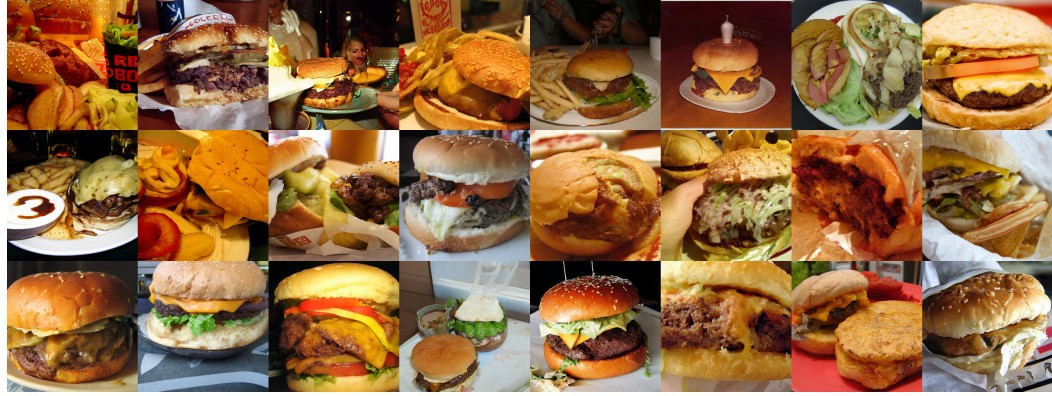

Figure 31: ImageNet 512x512 images of Class 933 (cheeseburger) generated using SiD$^2$A without classifier-free guidance, produced in a single generation step.

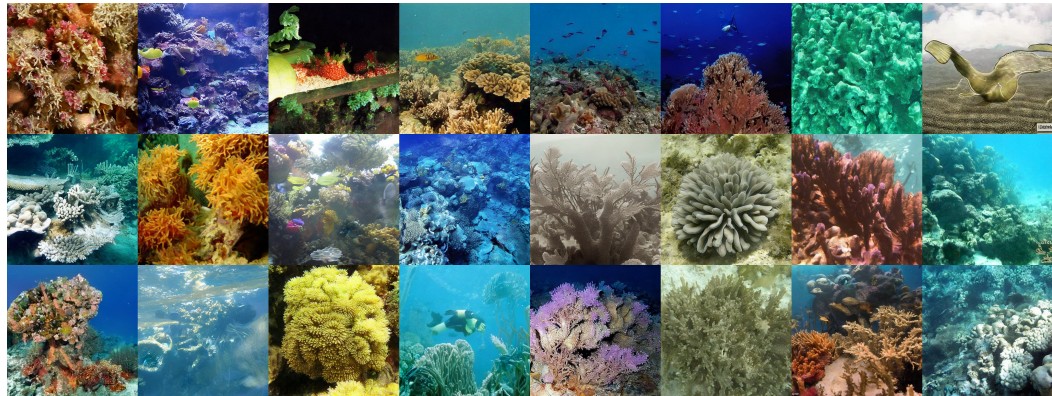

Figure 32: ImageNet 512x512 images of Class 973 (coral reef) generated using SiD$^2$A without classifier-free guidance, produced in a single generation step.

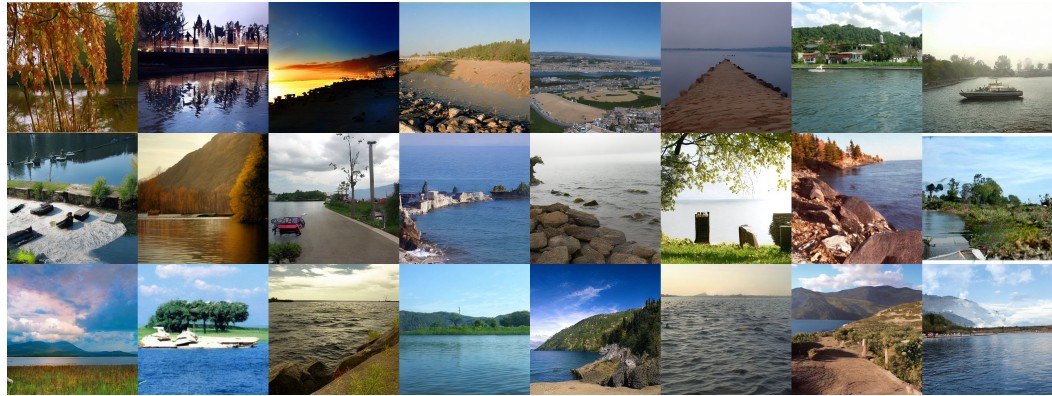

Figure 33: ImageNet 512x512 images of Class 975 (lakeside) generated using SiD$^2$A without classifier-free guidance, produced in a single generation step.

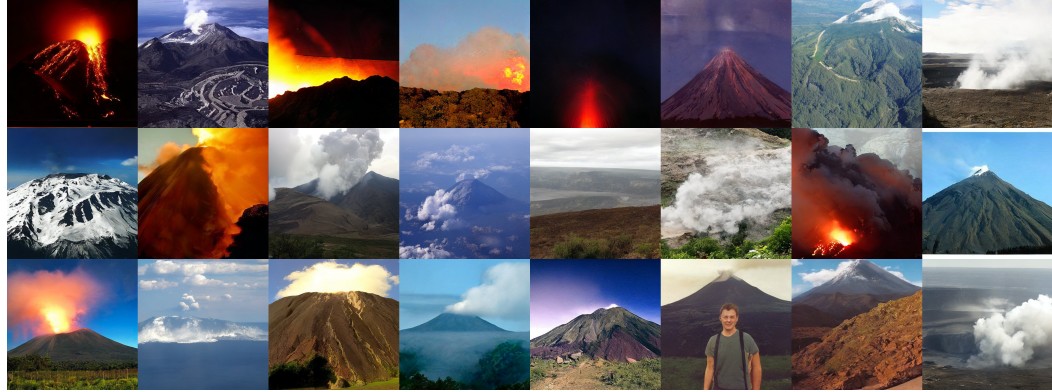

Figure 34: ImageNet 512x512 images of Class 980 (volcano) generated using SiD$^2$A without classifier-free guidance, produced in a single generation step.

