# OpenReview forum: "Adversarial Score identity Distillation: Rapidly Surpassing the Teacher in One Step"
_ICLR.cc/2025/Conference — ICLR 2025 Poster_

### Official Review · Reviewer_sPNq · 2024-11-02

**Soundness:** 3
**Presentation:** 3
**Contribution:** 3
**Rating:** 6
**Confidence:** 4

**Summary:**

This paper proposes an intuitive method, which mainly focuses on the improvement of score identity distillation method by introducing additional real images for adversarial training. The improvement is two fold. First, additional real images can boost generation performance over data-free score identity distillation by discrimating real and synthetic images. Second, it achieves faster convergence rate relative to the baselines. The experimental results on small-scale dataset support the superority of the proposed method.

**Strengths:**

+ This paper proposes an intuitive extension of score identity distillation and effectively improves the generation performance on small-scale datasets, such as CIFAR and ImageNet 64x64.
+ The proposed method achieves better performance than the main baseline, SiD and other diffusion methods.

**Weaknesses:**

+ More state-of-the-art generative methods are missing, such as DiT, RDM and MAR.
+ The method only conducts experiments on small-scale datasets or low-resolution images. I'm concern about the effectiveness of this method on large-scale dataset with high-resolution images. More results on large-scale datasets with high-resolution images consolidate the effectiveness of this method.
+ OSGAN[R1] adopts an one-step adversarial training on generative task. The comparion with OSGAN is needed to distinguish their differences and highlight the contributions of this paper.
+ This paper doesn't mention any plan about open source of code and trained weights. In spite of encouraging performance, I'm concern about the reproducibility of this paper.
+ Some motivations are not well clarified in the paper. See Questions section.
+ Some unnecessary equations, such as Eq between 149 and 151, can be removed.

[R1] Training Generative Adversarial Networks in One Stage. CVPR 2021.

**Questions:**

+ Why can incorporating average "fakeness" within each GPU batch into each pixel's loss solve the gap between diffusion distillation and adversarial learning?
+ Why does the proposed method can achieve faster convergence than baseline when trained from scratch?

---

> ### Author Response · Authors · 2024-11-15
> **Response to Reviewer sPNq**
>
> We would like to thank the reviewer for providing a clear summary of the improvements of SiDA over previous methods. Regarding the weaknesses pointed out, we would like to provide the following clarifications:
>
> **W1:** We will include a comprehensive comparison on ImageNet 512x512 with a wide range of baselines, including DiT and MAR, in our revision. For instance, using a 481M model with 256 autoregressive steps, MAR (Li et al., 2024) achieves an FID of 2.74 without CFG and using 100 diffusion NFEs, and an FID of 1.73 with CFG and using 100*2 diffusion NFEs. SiD$^2$A on EDM2-S (280M) already outperforms this, achieving an FID of 1.669 without CFG and with only a single generation step. Could you clarify which paper you are referring to for RDM?
>
>
>
> **W2:** We will add ImageNet 512x512 results and detailed comparisons with existing methods in the revised paper. Please refer to our general response for the current results on this point.
>
> **W3:** The term “One-Step” is defined differently in OSGAN and SiDA. In SiDA, we retain the alternating training between the discriminator (the encoder of the fake score network) and generator, referring to the single forward pass of the generator as “One-Step” generation. OSGAN, however, modifies alternating training into “One-Step” training, eliminating the need for alternation. We will clarify this distinction in the revision.
>
> **W4:** We commit to open-sourcing both the PyTorch code and model checkpoints. These are not provided at the moment to prevent any accidental violations of ICLR's anonymity requirements.
>
> **Q1:** A gap exists between the pretrained teacher's estimated score and the true score. The addition of an adversarial loss helps bridge this gap, guiding the generator toward a closer approximation of the true data distribution.
>
> **Q2**: In SiD, the generator is guided by a biased teacher network, while in SiDA, this bias can be corrected with the help of an adversarial loss that leverages real data. Even when using as little as 5% of the real data, this real-data-based guidance remains highly effective. Please refer to our response to Reviewer kYRQ's Q1 for further details.

---

> > ### Comment · Reviewer_sPNq · 2024-11-26
> >
> > + RDM refers to "Return of Unconditional Generation: A Self-supervised Representation Generation Method".
> > + The comparative experiments about the listed SOTA methods are required to supplement in the main body of paper. Moreover, the performance comparison on ImageNet 256x256 is a more popular setting. I would like to see more results on the experiment of ImageNet 256x256, which can well consolidate the effetiveness of this paper.
> > + The explaination about "incorporating average fakeness within each GPU batch" is still not clearly enough for me. I can not well capture the motivation of this design. I recommend that the authors can provide deeper and more detailed clarification about this point.

---

> > > ### Author Response · Authors · 2024-11-26
> > >
> > > We greatly appreciate the reviewer for providing additional feedback.
> > >
> > > 1. Thank you for suggesting RDM (Return of Unconditional Generation: A Self-supervised Representation Generation Method) as an additional reference. We have carefully reviewed RDM, also referred to as RCG in the paper, and believe its contribution is orthogonal to ours.
> > >
> > >      - RDM's primary focus is not on proposing a new diffusion training, inference, or distillation algorithm but on developing a label-free method to incorporate guidance into existing diffusion models, such as LDM, ADM, and DiT, to enhance their unconditional generation performance.
> > >
> > >      - Notably, SiDA does not currently use CFG, making it highly plausible that RDM could be integrated with SiDA to further improve its unconditional generation performance. However, RDM was primarily evaluated on ImageNet 256×256, while SiDA was evaluated on ImageNet 512×512, making the results not directly comparable. On CIFAR-10 (unconditional), RDM achieved an FID of 2.62, which is worse than SiD, SiDA, and SiD$^2$A, with FIDs of 1.923, 1.516, and 1.499, respectively.
> > >
> > >     - For these reasons, we view RDM as complementary to SiDA and do not consider a direct comparison necessary for this paper.
> > >
> > > 2. We have relocated that comprehensive comparison table on ImageNet 512x512 to the main body of the paper as Table 5.
> > >
> > >     - While both ImageNet 256×256 and ImageNet 512×512 are popular benchmarks for validating high-resolution performance, ImageNet 512×512 has seen more frequent use in recent papers published or preprinted in 2023 and 2024.
> > >
> > >    - Our experiments on ImageNet 512x512 compare over 25 different methods (with more than 40 settings in total) on ImageNet 512×512, making them comprehensive.
> > >
> > >    -  Although providing results on ImageNet 256×256 could offer additional insights, we believe its contribution to the paper's overall quality and impact is limited. Additionally, to the best of our knowledge, pretrained EDM2 model checkpoints for ImageNet 256×256 are not publicly available, and pretraining such a checkpoint would require a substantial investment of both human effort and computational resources. To facilitate further research, we will release our code, enabling those with the necessary resources to distill their pretrained diffusion models on ImageNet 256×256 and share the resulting checkpoints with the community. For downstream tasks requiring ImageNet 256×256, we recommend generating images at 512×512 using our methods and downsampling them to the desired resolution.
> > >
> > > 3. We have added a sentence immediately following Equation 9 to further clarify our design: While the generator loss $\mathcal L_{\theta,t}^{(\text{sid})}$ is influenced by the gap between the teacher network $\phi$ and its optimal value $\phi^*$, the added  term
> > > $\mathcal L_{\theta}^{(\text{adv})}$
> > > directly interacts with the training data to help bridge this gap.
> > >
> > >     - The key insight is that addressing the teacher's bias can significantly enhance the convergence speed and performance of SiD, a state-of-the-art data-free distillation method.
> > >
> > >      -   In this paper, we integrated per-GPU batch-average fakeness into the SiD loss, adhering to the original loss reweighting strategy of SiD, and found it to be highly effective. However, we acknowledge that alternative ways to incorporate adversarial loss, potentially working as effectively or even more effectively with an imperfect teacher, are worth exploring in future research.
> > >
> > > We hope these clarifications further highlight the contributions of the paper. Please feel free to reach out with any additional questions or suggestions.

---

### Official Review · Reviewer_4cAk · 2024-11-04

**Soundness:** 3
**Presentation:** 2
**Contribution:** 2
**Rating:** 5
**Confidence:** 4

**Summary:**

This paper introduces an improved version of score identity distillation and proposes training SiD with an adversarial objective.
The main motivation is that the lack of real data in the original training of SiD can potentially lead to discrepancies to the original data distribution. The authors propose to reuse the encoder part of the fake score model as the backbone of discriminator, and use its predicted realness and fakeness scores to improve the generation results further, at no cost of additional model parameters. By combining SiD with adversarial training, together with additional techniques such as initializing the student model with the SiD pretrained model, the proposed method achieved improved one-step diffusion results on multiple datasets.

**Strengths:**

- The discussions in the experiment section are very comprehensive. The detailed analysis of the training statistics across the training steps is appreciated.

- The paper presented the background such as SiD and adversarial loss in diffusion well.

**Weaknesses:**

- Presentation

In the technical parts of the paper, which is mainly the Section 4. I frequently find the presentations not clear enough. My biggest confusion arises from the 'Loss Construction that Respects the Weighting of Diffusion Distillation Loss' paragraph. I understand that balancing loss terms can be tricky, and I acknowledge that this is an important topic to discuss. But from what the author presents, I didn't find how the proposed weight 'honors the existing weighting mechanisms' and, more importantly, automatically balances the loss regardless of the datasets. I read the Eq 7 and Eq 8 and found nothing particularly new. How is Eq 7 fundamentally different from PatchGAN?

- Novelty

Although the paper targets a problem of SiD that is not previously discussed, the solution of adding an adversarial term is by no means a novelty under the context of diffusion distillation. The adversarial object has been proven very effective in many works, such as UFOGen and SD3-turbo. The proposed method here is a simple mixture of two existing methods.

- Evaluation and performance

Most of the experiment settings used in this paper are small-scale. And the authors only compared the proposed method with very few other methods in the 512x512 ImageNet experiment. Since efficiency is one of the main advantages the authors are trying to advocate, it is necessary to include larger-scale evaluations, such as text-to-image experiments, to allow for fair comparisons with recent methods.

I also have a slight concern about the advantage of efficiency. It is true that, according to Figure 3, the FID of the proposed method decreases faster compared to the baseline SiD in the later half of the training; I'd like to point out that this is achieved with an additional loss term, meaning each iteration probably optimizes the model with a larger gradient magnitude. Also, the adversarial loss term will inevitably introduce more computation, so the model is not necessarily able to finish training faster compared to SiD.

**Questions:**

Please see the Weakness section.

---

> ### Author Response · Authors · 2024-11-15
> **Response to Reviewer 4cAk**
>
> We thank the reviewer for providing a clear summary of our paper.
>
> ### **Presentation Clarification**
> We would like to clarify our statement that “the proposed weight honors the existing weighting mechanisms.” In the training of the generator and fake score networks in SiDA, we adopt the weighting strategies from SiD, which have been proven to be highly effective and stable. These strategies assign weights to diffused images based on their signal-to-noise ratios. When incorporating the adversarial loss, our goal was to minimize disruption to these established weighting mechanisms while leveraging the adversarial loss to correct the bias of the pretrained teacher.
>
> To achieve this, we average the spatial outputs of the encoder across all locations in a batch, scale this average by a constant, and add it to the unweighted SiD loss for the generator and the unweighted $x_0$-prediction denoising loss for the fake score network. This approach has proven effective across four datasets using EDM without dataset-specific tuning and six model sizes (ranging from 125M to 1.5B parameters) when distilling EDM2 on ImageNet 512x512 without model-specific tuning.
>
> The key takeaway in Eqs. (7–10) is that SiDA’s weighting mechanism remains unaffected by the introduction of the batch-averaged adversarial loss. Additionally, Eq. (7) differs from PatchGAN in several significant ways: (1) the input is noise-corrupted and depends on $t$; (2) it is reweighted in Eq. (8) using data-specific weights; and (3) the logits of the discriminator are derived from the output of the encoder component of the diffusion U-Net after channel-wise pooling.
>
> ### **Novelty**
> As discussed in *Related Work*, this paper is motivated by the success of Truncated Diffusion, Diffusion-GAN, and SiD. The novel contribution lies in how we combine the SiD loss with the Diffusion-GAN loss without introducing additional parameters beyond those in SiD while preserving its weighting mechanism.
>
> Existing adversarial diffusion distillation methods often rely heavily on their adversarial components for good performance. For instance, Adversarial Diffusion Distillation (ADD) by Sauer et al. (2023) barely outperforms using adversarial loss alone and performs poorly without it. In contrast, SiDA builds upon the already strong baseline of SiD, with adversarial loss added specifically to correct the bias of the pretrained teacher network. This novel insight into correcting the pretrained teacher’s bias using adversarial loss adds significant value to the field and highlights the importance of this work, making it a compelling contribution worthy of publication.
>
> ### **Evaluation and Performance**
> We will expand our evaluation to include distillation results for EDM2 across six model sizes (125M to 1.5B parameters), pretrained on ImageNet 512x512. Additionally, we will compare SiDA to over 25 baselines that report results on ImageNet 512x512. The results show that SiDA-distilled EDM2 achieves lower FIDs despite being smaller, significantly faster, and not relying on CFG.
>
> Regarding text-to-image experiments, we have observed promising results with SiDA. However, we believe these results merit their own publication, focusing not only on achieving lower FID but also on improving image-text alignment, which is beyond the scope of this paper.
>
> ### **Efficiency**
> We will provide further comparisons demonstrating that SiDA is only about 10% slower per iteration than SiD. Despite this slight overhead, SiDA achieves significantly faster FID reduction and surpasses the teacher in less time, making it approximately an order of magnitude faster than SiD when considering its efficiency in improving performance.

---

> > ### Author Response · Authors · 2024-11-27
> >
> > Dear Reviewer 4cAk,
> >
> > We would be grateful if you could let us know if there are any additional questions we can address. We eagerly await your feedback on our responses.
> >
> > Thank you,
> >
> > The Authors of SiDA

---

### Official Review · Reviewer_kYRQ · 2024-11-04

**Soundness:** 3
**Presentation:** 2
**Contribution:** 3
**Rating:** 8
**Confidence:** 4

**Summary:**

The paper introduces Score Identity Distillation with Adversarial loss. The method aims to distill a teacher diffusion model to reduce the number of inference steps to only one. The primary contribution is the combination of SiD with the use of real images and adversarial objective, which not only reduces the number of required inference steps but also improves generation quality, achieving a lower FID score across Cifar10, ImageNet-64, FFHQ-64 and AFHQ-64.

**Strengths:**

- Achieves state-of-the-art performance with only one inference step
- The methods demonstrates improvement over the teacher network thanks to the real images
- Training remains relatively stable despite the use of an adversarial loss

**Weaknesses:**

- Requires real images for training, which is not necessary for other distillation methods
- Experiments are demonstrated only on very small images resolution
- Ablation studies are limited to the parameter $\alpha$
- A schematic of the framework would greatly improve the paper's readability.

**Questions:**

- How does the method perform as the number of available real images changes? What happen if you use only 10% of the training real images?
- Why not apply latent diffusion for larger images? Using VAE to compress the image would help to see if the method can scale to bigger images, without increasing the training time
- Does the approach generalize to teacher models other than VP-EDM?
- The paper arbitrarily sets $\lambda_{sid}$ = 100 and $\lambda_{adv}$ = 0.01 without discussion or ablation, although these hyperparameters likely play an important role in the method. How do these values influence the performance metrics?
- In Figure 1, for $\alpha$=1.2 (the yellow line), after approximately 1000 iterations, the Inception Score (IS) drops and the FID increases sharply; additionally, SiDA with $\alpha$=1.2 is absent from Table 1. Does this imply that the model fails to converge?

---

> ### Author Response · Authors · 2024-11-14
> **Response to Reviewer kYRQ**
>
> We appreciate the reviewer's positive feedback on our paper.
>
> - In response to the identified weaknesses, we provide the following clarifications:
>
>    - **W1:** We’d like to clarify that data-free distillation, which does not require real or teacher-synthesized images, is a distinctive feature of methods like Diff-Instruct and SwiftBrush, which optimize KL divergence under diffusion, and SiD, which optimizes Fisher divergence under diffusion. In scenarios lacking real data, SiD stands as the state-of-the-art choice. Most other distillation methods, including consistency distillation, CTM, and DMD, necessitate access to real or teacher-synthesized data. SiDA not only surpasses the teacher model but also outperforms both data-free and non-data-free distillation methods.
>
>   - **W2:** We will include results on distilling EDM2 pretrained on ImageNet 512×512 in the revised paper.
>
>   - **W3:** The revised paper will detail our approach to model parameter selection. Given that SiDA demonstrates robustness to these settings, we believe the provided guidelines are sufficient. However, we are open to conducting additional ablation studies if the reviewer deems them necessary for a deeper understanding of our method.
>
>   - **W4:** A schematic illustration of the SiDA algorithm will be added to the paper.
>
> - Addressing the reviewer's specific questions:
>
>   - **Q1:** The VAE latents for the ImageNet (512×512) dataset total approximately 157 GB. In our experiments with SiDA for EDM2-XS, we tested utilizing only 5% of the images (7.8 GB). The Appendix includes a comparison of the FID trajectories between using 5% and the full dataset. The results indicate that, with just 5% of the data, the SiDA framework achieves an initial convergence rate closely following that of the full dataset. However, it reaches a plateau earlier and converges to a higher FID—worse than using the full data but still clearly outperforming the teacher. This finding actually suggests potential applications of SiDA in domain adaptation, which we plan to explore in future research.
>
>   - **Q2:** We have implemented the suggested changes; please refer to our response in W2.
>
>   - **Q3:** Please see below for a separate response.
>
>   - **Q4:** The revised paper includes a description of our parameter selection process. The approach involves adjusting parameters so that the SiD loss and adversarial loss components have comparable scales for both the generator and fake score networks, maintaining values roughly between 1 and 10,000 to prevent underflow or overflow in fp16. Notably, setting the adversarial loss coefficient too low results in SiDA's performance aligning closely with SiD, as the adversarial loss becomes dominated by the SiD loss component.
>
>   - **Q5:** In our ablation study on CIFAR10 (unconditional), we observed that the current SiDA implementation with $\alpha=1.2$ can diverge. This issue can be mitigated by extending the initial warmup period using the SiD loss alone before introducing the adversarial loss. For SiD$^2$A, which is essentially SiDA with the generator initialized from a much better starting point, the model performs well at $\alpha=1.2$ and remains stable even when $\alpha$ is increased up to 1.5.
>
> We trust these clarifications address your concerns and welcome any further discussion.

---

### Official Review · Reviewer_Ajww · 2024-11-08

**Soundness:** 3
**Presentation:** 2
**Contribution:** 3
**Rating:** 6
**Confidence:** 3

**Summary:**

The authors propose SiDA (Score identity Distillation with Adversarial Loss), an enhanced version of Score identity Distillation (SiD) that aims to improve diffusion model training efficiency. The key innovation is the introduction of an adversarial loss that helps distinguish between real and generated latents at different noise levels. By reusing the fake score network in the original SiD, the authors propose a discriminator design that utilizes the encoder feature map to obtain the discriminator score. The proposed method achieves SOTA results on CIFAR-10, ImageNet-64, and other datasets.

**Strengths:**

-	The authors repurpose the fake score network as an additional discriminator with direct using the average of the encoder features as the discriminator score. This design introduces the adversarial objective to the original SiD with minimal additional parameter cost.
-	The proposed method SiDA significantly outperforms SiD and achieves superior 1-step generation performance on various datasets.

**Weaknesses:**

-	The discriminator design is heavily based on the U-Net encoder-decoder architecture. It is not straightforward to generalize to transformer-based diffusion models, e.g., DiT, where there is no clear partition of the encoder and decoder.
-	The current submission lacks the evaluation of Diffusion models on higher resolution, e.g., ImageNet 256x256. The cost overhead from both discriminator training and maintaining an additional fake score network can be exaggerated on high-resolution data.

**Questions:**

-	Why $L^1(\theta)$ is still needed with the additional discriminator, i.e., why SiDA still needs tuning $\alpha$?

---

> ### Author Response · Authors · 2024-11-14
> **Response to Reviewer Ajww**
>
> We appreciate the reviewer’s recognition of SiDA’s SOTA performance.
>
>
> 1. *Can SiDA be used to distill DiT-based models?*
>
>
>     **Response:** Our research primarily focuses on distilling U-Net-based EDM models, which are well-recognized and benchmarked in diffusion-based generation and distillation. We have now also added results for EDM2, which shows a significantly lower FID on ImageNet 512 x 512 compared to all existing DiT-based models, as detailed in Table 2 of the EDM2 paper and our forthcoming revision.
>
>
>      Despite this, we acknowledge the versatility of DiT-based architectures. SiDA is not inherently limited to U-Net backbones and shows potential for adapting to transformer-based architectures. For example, in the context of transformer-based diffusion models, we could divide the architecture into “encoder” and “decoder” components. Insights from studies like REPA (Yu, Sihyun, Sangkyung Kwak, Huiwon Jang, Jongheon Jeong, Jonathan Huang, Jinwoo Shin, and Saining Xie. "Representation Alignment for Generation: Training Diffusion Transformers Is Easier Than You Think." arXiv preprint arXiv:2410.06940 (2024).) suggest that aligning self-supervised features from the encoder, particularly those from middle layers (6th to 8th in a 24-layer model), with a diffusion model could enhance DiT/SiT model training.
>
>
>     While integrating SiDA with transformer-based models is intriguing, this exploration is beyond the scope of our current research. We plan to pursue this direction in future projects and invite researchers with the necessary computing resources to adapt our SiDA code, which will be open-sourced, to distill DiT-based models and contribute their findings to the community.
>
>
> 2. *Can SiDA be applied to high-resolution models?*
>
>
>      **Response:** Yes, SiDA excels in high-resolution settings, achieving remarkably low FIDs on ImageNet 512x512 without CFG and through one-step generation. We have documented our current results for distilling EDM2 and will provide further updates after revising our paper based on the reviewers' feedback.
>
>
> 3. *Why is $L^1(\theta)$ still necessary with an additional discriminator, i.e., why does SiDA still require tuning $\alpha$?*
>
>
>      **Response:** Omitting $L^1(\theta)$, or setting $\alpha=0$, was feasible in SiD but resulted in unsatisfactory FID scores. With the inclusion of adversarial loss, this configuration in both SiDA and SiD$^2$A now yields satisfactory outcomes, yet still underperforms compared to the recommended settings of $\alpha=1.0$ for SiDA and $\alpha=1.0$ or $1.2$ for SiD$^2$A. This indicates that, despite the support from adversarial loss, maintaining $\alpha>0$ is important to correct the approximation bias in the SiD gradient effectively.

---

> > ### Comment · Reviewer_Ajww · 2024-12-02
> > **Thanks for the response**
> >
> > Thanks to the authors for the response. All of my concerns have been alleviated. I will retain my positive view of the submission.

---

> > > ### Author Response · Authors · 2024-12-02
> > >
> > > Thank you for affirming your positive evaluation of our submission. Please do not hesitate to reach out if you have any further questions or comments.

---

### Author Response · Authors · 2024-11-13
**Preliminary Response (Updated): Addressing Concerns on High-Resolution Image Results**

Dear Reviewers,

We appreciate your valuable feedback. While we are preparing a comprehensive revision, we would like to promptly address the shared concern regarding the absence of results for high-resolution images. To this end, we have adapted the SiDA codebase to distill EDM2 models of various sizes pretrained on ImageNet 512×512, yielding the following results:

| **Method** | **CFG** | **NFE** | **XS (125M)** | **S (280M)** | **M (498M)** | **L (777M)** | **XL (1.1B)** | **XXL (1.5B)** |
|------------|---------|---------|---------------|--------------|--------------|--------------|---------------|----------------|
| EDM2       | N       | 63      | 3.53          | 2.56         | 2.25         | 2.06         | 1.96          | 1.91           |
| EDM2       | Y       | 63x2      | 2.91          | 2.23         | 2.01         | 1.88         | 1.85          | 1.81           |
| sCT        | Y       | 1       | -             | 10.13        | 5.84         | 5.15         | 4.33          | 4.29           |
| sCT        | Y       | 2       | -             | 9.86         | 5.53         | 4.65         | 3.73          | 3.76           |
| sCD        | Y       | 1       | -             | 3.07         | 2.75         | 2.55         | 2.40          | 2.28           |
| sCD        | Y       | 2       | -             | 2.50         | 2.26         | 2.04         | 1.93          | 1.88           |
| SiD        | N       | 1       | 3.353 ± 0.041 | 2.707 ± 0.054| 2.060 ± 0.038| 1.907 ± 0.016| 1.888 ± 0.030 | 1.969 ± 0.029              |
| SiDA       | N       | 1       | 2.228 ± 0.037 | 1.756 ± 0.023| 1.546 ± 0.023| 1.501 ± 0.024| 1.428 ± 0.018 | 1.503 ± 0.020             |
| SiD²A      | N       | 1       | **2.156 ± 0.028** | **1.669 ± 0.019** | **1.488 ± 0.038** | **1.413 ± 0.022** | **1.379 ± 0.017** | **1.366** ± 0.015              |


The results for EDM2 are sourced from Karras et al. (2024), while those for sCT and sCD are obtained from Lu and Song (2024). Although Lu and Song's work is concurrent with ours, we include their results here to highlight the superior performance of SiDA.

We observe that SiD²A achieves an FID of 1.366 on EDM2-XXL, marking the lowest score reported to date. This result is notable as it is obtained without classifier-free guidance (CFG), post-hoc exponential moving average (EMA), and utilizes only a single generation step. Furthermore, both SiDA and SiD²A applied to EDM2-S (280M parameters) surpass the performance of the largest teacher model, EDM2-XXL (1.5B parameters), as well as sCD and sCT methods. We also note that SiD, the baseline model adapted by us for EDM2, achieves performance comparable to that of the teacher model. This observation aligns with previous findings on smaller datasets and EDM models, demonstrating SiD's effectiveness across varying scales.

**References:**

- Tero Karras, Miika Aittala, Jaakko Lehtinen, Janne Hellsten, Timo Aila, and Samuli Laine. "Analyzing and Improving the Training Dynamics of Diffusion Models." In *Proceedings of the IEEE/CVF Conference on Computer Vision and Pattern Recognition*, pp. 24174–24184, 2024.

- Cheng Lu and Yang Song. "Simplifying, Stabilizing and Scaling Continuous-Time Consistency Models." *arXiv preprint arXiv:2410.11081*, 2024.

We will provide a more detailed response and a revised paper in due course.

---

> ### Author Response · Authors · 2024-11-19
> **Request for Review of Our Revised Paper and Response**
>
> Dear Reviewers,
>
> Thank you for your valuable feedback, which has guided our revisions to the manuscript. We have incorporated all the revisions outlined in our responses to each review. We will soon further update the paper to include visualizations that demonstrate how image generation rapidly improves its visual quality during the distillation of EDM2 on ImageNet 512x512. Additionally, we will showcase randomly generated images from specific class labels—if there are particular classes you would like to see, please let us know.
>
> We invite you to review our updated paper and our responses. We appreciate any further feedback you may provide.
>
> Thank you,
>
> Authors of SiDA

---

> ### Author Response · Authors · 2024-11-20
> **Visualization of ImageNet 512x512 Random Generations are now available in the revised paper**
>
> We have completed the revision of the manuscript, incorporating class-conditional random generations of ImageNet 512x512 using the same set of random noise in Figures 16–34. Additionally, we have added Figures 8-10 to illustrate the rapid improvement of SiD, SiDA, and SiD$^2$A in visual quality as the iterations progress.

---

### Meta-Review · Area_Chair_n45c · 2024-12-20

**Metareview:**

This paper improves score identity distillation by integrating real data with adversarial loss, which enhances generation quality even surpassing the teacher in one-step generation, while also improving distillation efficiency. Overall, this work receives positive scores, and the reviewers think the idea of introducing real data is interesting, the method is well designed, and the performance gain is significant. Some concerns include: 1) the method focuses only on U-Net without considering DiT, 2) most experiments are small-scale; and 3) the use of adversarial loss is not new. AC agrees this is an interesting advance on score identity distillation, and recommends acceptance as poster.

**Additional Comments On Reviewer Discussion:**

Some concerns raised by the reviewers:

1) the method focuses only on U-Net without considering DiT;
2) most experiments are small-scale;
3) the use of adversarial loss is not new.

During the rebuttal, the authors provided reasonable justifications for all the above issues, and included more experimental results. The reviewers did not raise further major concerns.

I am generally exciting about the idea and result that integrating real images during score distillation can yield generation quality better than the teacher.

---

### Decision · Program_Chairs · 2025-01-22

Accept (Poster)